# KNOWLEDGE-SENSITIVE DYNAMIC MODULE EDITING: PRECISE KNOWLEDGE REVISION FOR MULTIMODAL LARGE LANGUAGE MODELS

## ABSTRACT

Multimodal Large Language Models (MLLMs) struggle with efficient knowledge updates because their internal representations distribute information across lengthy and heterogeneous visual-textual sequences. This distribution makes traditional "locate-then-edit" methods, despite being highly effective in text-only models, largely ineffective for MLLMs. The resulting challenges include inaccurate localization of knowledge, poor generalization of edits, and unintended damage to unrelated knowledge. To bridge this gap, we introduce *KDKE*, a novel **K**nowledge-sensitive **D**ynamic multimodal **K**nowledge **E**diting framework tailored for MLLMs. *KDKE* introduces an Integrated Module Contribution Score to precisely quantify the impact of different modules on specific knowledge outputs. This enables a dynamic module selection mechanism that identifies critical parameters for each edit instance adaptively. We further develop a constrained adaptive editing algorithm, which injects LoRA parameters into selected modules and optimizes them under multi-objective constraints to ensure reliable editing, robust generalization, and strict locality. Extensive experiments on multiple model architectures and benchmarks demonstrate that *KDKE* superior editing accuracy and consistently strong overall performance, providing an effective and reliable solution for knowledge editing in multimodal settings.

## 1 INTRODUCTION

Multimodal Large Language Models (MLLMs) have rapidly advanced to become a cornerstone of artificial intelligence, demonstrating strong capabilities in comprehending and generating cross-modal content that integrates both visual and textual information (Park & Kim, 2023; Alayrac et al., 2022; Li et al., 2022). These models leverage Transformer architectures (Vaswani et al., 2023) to process fused visual and textual embeddings, aligning visual inputs with pre-trained linguistic representations. This enables a wide range of applications, from visual question answering to complex reasoning tasks (Wang et al., 2022; Wu et al., 2023). Despite these advancements, a significant challenge persists: how to efficiently and accurately update the knowledge embedded within MLLMs. The real world is constantly changing: facts evolve, new entities emerge, and errors in training data are continuously discovered. This dynamism renders full model retraining for every update computationally prohibitive and environmentally unsustainable. Consequently, model editing techniques (Wang et al., 2024; Yao et al., 2023) have emerged, aiming to precisely modify a model's behavior for specific knowledge while largely preserving its performance on unrelated tasks. In the context of pure-text Large Language Models (LLMs), the "locate-then-edit" paradigm (Meng et al., 2023a;b) has proven highly effective, which typically employs causal tracing to identify and rank-one adjust knowledge-storing modules.

However, directly adapting this paradigm to MLLMs remains challenging. In pure-text settings, knowledge is often structured in subject–relation–object triplets (e.g., 'Trump','is a president of', 'USA'). In MLLMs, by contrast, the specific subject is replaced by an image and samples like "<image> This photo shows a president of the USA". Knowledge representations are thus distributed both spatially and semantically across lengthy and heterogeneous visual sequences. This distribution makes it challenging to precisely locate where knowledge is stored within the model's parameters and to identify key knowledge anchors in the input that are crucial for effective editing.

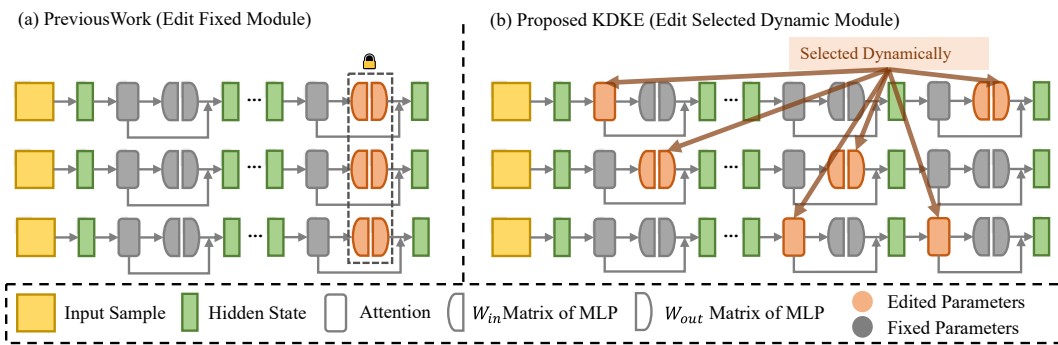

Figure 1: Comparison of conventional knowledge editing methods for local layers of models and the dynamic module selection method we proposed.

As a result, these complexities further complicate parameter editing in MLLMs. Although recent work (Basu et al., 2024) has attempted to adapt causal tracing to MLLMs via visual and textual constraints, these methods require manual dataset reconstruction and often yield imprecise localization across broad sets of layers, resulting in suboptimal editing performance. Therefore, two central questions remain: How can multimodal knowledge be accurately localized? And how can we achieve effective parameters editing in MLLMs?

To tackle these challenges, we propose **KDKE**, a knowledge-sensitive dynamic editing framework for MLLMs. Our approach is motivated by the observation that in LLMs, causal tracing-based attribution methods tend to localize knowledge to a specific, statistically consistent set of modules. Our assumption is that, in the multimodal setting, different types of knowledge are stored in distinct layers and modules. Verifying this using existing attribution techniques is prohibitively inefficient; methods such as Meng et al. (2023a) require computational costs far exceeding a single forward pass per edit. To efficiently attribute model predictions to specific modules, we introduce a *Integrated Module Contribution Score*, which quantifies the influence of each module (e.g., attention and FFN layers) on the final output token by integrating its direct probability output, normalized logit value, and support saliency relative to other candidates. This score serves as core evidence for our assumption and enables fine-grained, instance-aware attribution within a single inference step. Building on this efficient attribution, our framework incorporates a *Dynamic Module Selection* mechanism that adaptively identifies critical modules for each editing instance, along with the *Constrained Adaptive Editing* algorithm, which injects LoRA (Hu et al., 2021) parameters only into selected modules for localized fine-tuning. A multi-objective loss regularizes the process to ensure editing reliability, paraphrase generalization and minimal impact on base model performance.

Our contributions are as follows:

- We identify and address two underexplored yet critical challenges in MLLM knowledge editing: fine-grained localization of cross-modal knowledge and efficient instance-specific parameter modification. Unlike prior work that directly transplants text-based methods, we propose a tailored editing framework capable of handling the distributed and semantically entangled nature of multimodal representations.

- We present a novel dynamic editing paradigm that moves beyond static layer-wise modifications. Central to our approach is a lightweight, attribution-based module selection mechanism that dynamically identifies and edits critical components on a per-instance basis—significantly improving precision and efficiency compared to conventional causal tracing-based localization.

- We conduct extensive experiments on three popular MLLMs and two multimodal benchmarks. Results demonstrate our method achieves state-of-the-art performance on the majority of metrics and obtains the highest average score across all editing scenarios.

## 2 PRELIMINARY

### 2.1 MULTIMODAL LARGE LANGUAGE MODELS

Multimodal Large Language Models (MLLMs) follow the fundamental paradigm of autoregressive Large Language Models (LLMs). Specifically, such models map an input pair consisting of a visual input $\mathbf{x}_v$ and a text prompt $\mathbf{x}_t$ to a textual output $\mathbf{o}$ via a parametric function $f_\theta : \mathcal{X}_v \times \mathcal{X}_t \to \mathcal{O}$, i.e., $\mathbf{o} = f_\theta(\mathbf{x}_v, \mathbf{x}_t)$. The core internal transformer module can be formally represented as $\hat{f}_\theta : \mathcal{E}_v \times \mathcal{E}_t \to \mathcal{Y}$. This module transforms a fused embedding sequence $\varepsilon = \varepsilon_v \oplus \varepsilon_t \in \mathbb{R}^{N \times d_h}$ into a probability distribution $\mathbf{y} \in \mathcal{Y} \subset \mathbb{R}^{|\mathcal{V}|}$, which is used to predict the next textual token based on a predefined vocabulary $\mathcal{V}$. Here, $\varepsilon_v \in \mathcal{E}_v \subset \mathbb{R}^{N_v \times d_h}$ denotes the visual embedding sequence, and $\varepsilon_t \in \mathcal{E}_t \subset \mathbb{R}^{N_t \times d_h}$ refers to the text prompt embedding sequence. $N_v$, $N_t$, and $N$ correspond to the lengths of the visual embedding sequence, the text embedding sequence, and the total fused embedding sequence, respectively. $d_h$ represents the hidden dimension of the intermediate transformer layers. Within the transformer architecture of MLLMs, the hidden state at each layer progressively integrates multimodal information through residual connections. Here, $\mathbf{a}_n^l$ is the output of the self-attention mechanism, and $\mathbf{m}_n^l$ is the output of the feed-forward network. This mechanism ensures effective gradient propagation and preserves low-level features. The hidden state at each layer carries representations at different levels of abstraction: lower layers capture local features, while higher layers integrate global semantics. Although the final output is based on the top-layer hidden state, due to residual accumulation, all layers contribute directly or indirectly to the final prediction, enabling the gradual refinement and generation of cross-modal features.

### 2.2 MLLM KNOWLEDGE EDITING

Let $f_\theta$ be a Multimodal Large Language Model (MLLM), where $\theta$ denotes the model parameters. The model maps a visual input $\mathbf{x}_v$ and a textual input $\mathbf{x}_t$ to an output $\mathbf{o}$, i.e., $f_\theta(\mathbf{x}_v, \mathbf{x}_t) = \mathbf{o}$. Given an edit instance $(\mathbf{x}_v^e, \mathbf{x}_t^e, \mathbf{o}^e)$ such that $f_\theta(\mathbf{x}_v^e, \mathbf{x}_t^e) \neq \mathbf{o}^e$, the objective of a model editor $\mathcal{M}_E(\cdot)$ is to produce an edited model $f_{\theta_e} = \mathcal{M}_E(f_\theta, \mathbf{x}_v^e, \mathbf{x}_t^e, \mathbf{o}^e)$ that satisfies $f_{\theta_e}(\mathbf{x}_v^e, \mathbf{x}_t^e) = \mathbf{o}^e$, while adhering to the following criteria:

- **Reliability (Rel.):** The edited model should respond correctly to the precise edit instance:

$$\mathbb{E}_{(\mathbf{x}_v^e, \mathbf{x}_t^e, \mathbf{o}^e) \sim \mathcal{D}_e} \left[ \mathbb{I} \left\{ f_{\theta_e}(\mathbf{x}_v^e, \mathbf{x}_t^e) = \mathbf{o}^e \right\} \right] \tag{1}$$

- **Generalization (Gen.):** The edit should generalize to semantically equivalent variations of the edit instance. This includes:

  - **Textual Generalization (T-Gen.):**

$$\mathbb{E}_{(\mathbf{x}_v^e, \mathbf{x}_t^e, \mathbf{o}^e) \sim \mathcal{D}_e} \left[ \mathbb{E}_{\tilde{\mathbf{x}}_t^e \sim \mathcal{N}(\mathbf{x}_t^e)} \left[ \mathbb{I} \left\{ f_{\theta_e}(\mathbf{x}_v^e, \tilde{\mathbf{x}}_t^e) = \mathbf{o}^e \right\} \right] \right] \tag{2}$$

  - **Visual Generalization (I-Gen.):**

$$\mathbb{E}_{(\mathbf{x}_v^e, \mathbf{x}_t^e, \mathbf{o}^e) \sim \mathcal{D}_e} \left[ \mathbb{E}_{\tilde{\mathbf{x}}_v^e \sim \mathcal{N}(\mathbf{x}_v^e)} \left[ \mathbb{I} \left\{ f_{\theta_e}(\tilde{\mathbf{x}}_v^e, \mathbf{x}_t^e) = \mathbf{o}^e \right\} \right] \right] \tag{3}$$

- **Locality (Loc.):** The model should retain its original behavior on samples unrelated to the edit. This includes:

  - **Textual Locality (T-Loc.):**

$$\mathbb{E}_{(\mathbf{x}_v^e, \mathbf{x}_t^e, \mathbf{o}^e) \sim \mathcal{D}_e} \left[ \mathbb{E}_{(\emptyset, \hat{\mathbf{x}}_t^l) \sim \mathcal{U}(\mathbf{x}_t^l)} \left[ \mathbb{I} \left\{ f_{\theta_e}(\emptyset, \hat{\mathbf{x}}_t^l) = f_\theta(\emptyset, \hat{\mathbf{x}}_t^l) = \delta_t^l \right\} \right] \right] \tag{4}$$

  - **Multimodal Locality (I-Loc.):**

$$\mathbb{E}_{(\mathbf{x}_v^e, \mathbf{x}_t^e, \mathbf{o}^e) \sim \mathcal{D}_e} \left[ \mathbb{E}_{(\hat{\mathbf{x}}_v^l, \hat{\mathbf{x}}_t^l) \sim \mathcal{U}(\mathbf{x}_v^l, \mathbf{x}_t^l)} \left[ \mathbb{I} \left\{ f_{\theta_e}(\hat{\mathbf{x}}_v^l, \hat{\mathbf{x}}_t^l) = f_\theta(\hat{\mathbf{x}}_v^l, \hat{\mathbf{x}}_t^l) = \delta_v^l \right\} \right] \right] \tag{5}$$

where $\mathcal{D}_e$ is the distribution of edit instances, $\mathcal{N}(\cdot)$ denotes the neighborhood of semantically equivalent inputs, $\mathcal{U}(\cdot)$ represents the distribution of unrelated samples, and $\mathbb{I}\{\cdot\}$ is the indicator function.

## 3 METHOD

This section first defines the Integrated Module Contribution Score, which analyzes the contribution of different layers to the model's final output (Section 3.1). Based on the analysis in 3.1, we propose a Knowledge-sensitive Dynamic multimodal Knowledge Editing framework named **KDKE** (Section 3.2). Specifically, we design a dynamic module selection algorithm and perform local fine-tuning by injecting LoRA modules into the selected layers.

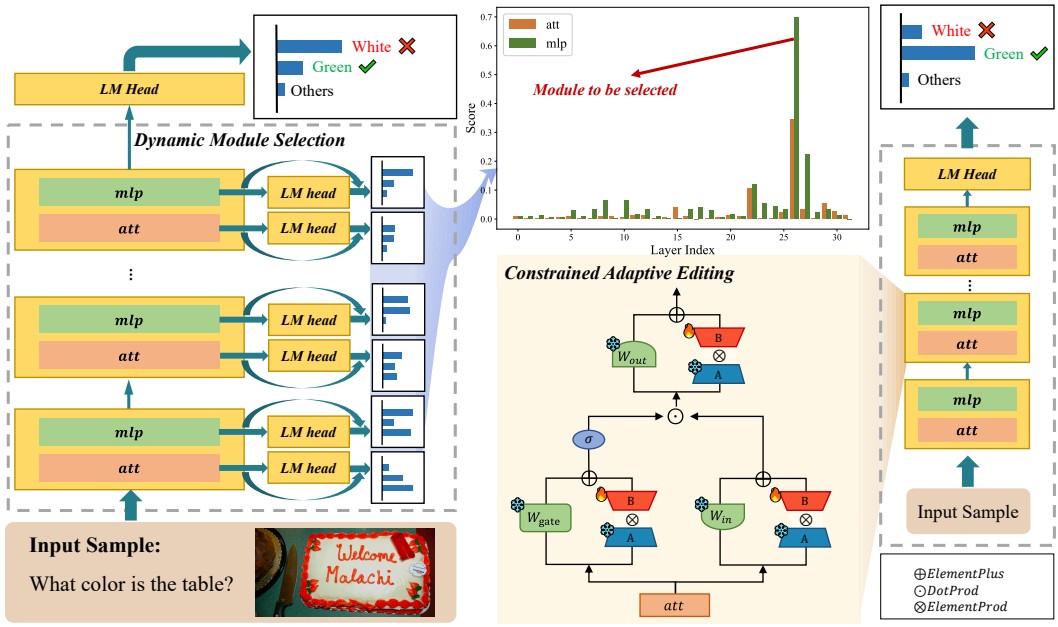

Figure 2: The proposed algorithm KDKE calculates the Integrated Module Contribution Score of outputs in each layer and module for a specific data input through a single inference calculation model, dynamically selects several modules with high scores, and adds Constrained Adaptive Editing for knowledge editing, so as to achieve precise modification of specific knowledge.

### 3.1 INTEGRATED MODULE CONTRIBUTION SCORE

Each token state in a transformer is part of the residual stream that all attention and MLP modules read from and write to (Elhage et al., 2021), the hidden state of the last token in the final layer is denoted as $h^L_{[N]}$, where

$$h^l_{[N]} = a^l_{[N]} + m^l_{[N]} + h^{l-1}_{[N]} \tag{6}$$

$$a^l_{[N]} = attn(h^{l-1}_{[0]}, h^{l-1}_{[1]}, ... h^{l-1}_{[N]}) \tag{7}$$

$$m^l_{[N]} = W^l_{out}\sigma(W^l_{in}(h^{l-1}_{[N]} + a^l_{[N]})) \tag{8}$$

$h^L_{[N]}$ is transformed through a linear projection $W_V$ followed by a softmax operation to obtain the probability distribution over the next token, i.e., $p = \text{softmax}(h^L_{[N]} W_V)$. Expanding equation 6, we derive:

$$h^L_{[N]} = h^0_{[N]} + \sum_{l=1}^{L} a^l_{[N]} + \sum_{l=1}^{L} m^l_{[N]} \tag{9}$$

Multiplying both sides by $W_V$ yields:

$$h^L_{[N]} W_V = h^0_{[N]} W_V + \sum_{l=1}^{L} a^l_{[N]} W_V + \sum_{l=1}^{L} m^l_{[N]} W_V \tag{10}$$

Given that the softmax function is monotonic, we observe that the token prediction distribution depends on the outputs of all attention and MLP modules across layers. As the same of observation with Chen et al. (2025b), we define a unified contribution score for each module as:

$$C_{[t]}(z) = \left[C_{[t]}^P(z)\right]^m \cdot \left[\eta \cdot C_{[t]}^O(z) + (1-\eta) \cdot C_{[t]}^S(z)\right]^{1-m} \tag{11}$$

where $z$ denotes the hidden representation output by the module, $m$ and $\eta$ are hyperparameters. Here, $C_t^P(z)$ represents the mapping probability of the module output:

$$C_{[t]}^P(z) = \text{softmax}(zW_V)_{[t]} \tag{12}$$

If the logit value output by a module is small, even if the resulting probability $C_t^P(z)$ after softmax is high, it might be due to the generally small logits of all other tokens, leading to a relatively high probability. However, its actual contribution to the final logit sum $h_N^L W_V$ could be negligible. To address this, we introduce a normalized logit value $C_t^O(z)$:

$$C_{[t]}^O(z) = \frac{(zW_V)_{[t]}}{\max_{l=1}^L \left(|(a_{[t]}^l W_V)_{[t]}|, |(m_N^l W_V)_{[t]}|\right)} \tag{13}$$

Furthermore, a module might assign a high mapping probability to the target token $t$, but if it also assigns similarly high probabilities to many other tokens, its "definitive support" for $t$ is relatively weak. Conversely, if a module assigns a moderate score to $t$ but very low scores to all other tokens, its contribution to $t$ is more "unique". To capture this, we first compute the gap $D_{[t]}(z)$ between the logit of $t$ and that of the second most likely token:

$$D_{[t]}(z) = (zW_V)_{[t]} - \max_{k \in \mathcal{V} \setminus \{[t]\}} (zW_V)_k \tag{14}$$

We then normalize this difference and ensure non-negativity, defining the Significant Contribution $C_{[t]}^S(z)$ as:

$$C_{[t]}^S(z) = \frac{\text{ReLU}(D_{[t]}(z))}{\max_{l=1}^L \max \left(\text{ReLU}(D_{[t]}(a_{[t]}^l)), \text{ReLU}(D_{[t]}(m_{[t]}^l))\right) + \epsilon} \tag{15}$$

Proposed Multi-level Module Contribution Score effectively captures the representations of the last input token across different layers and modules, enabling accurate distinction of module contributions in edited examples. The relevant theories proofs and experimental analysis are provided in the appendix B.

## 3.2 Knowledge-sensitive Dynamic Knowledge Editing

Building upon the computation method described in Section 3.1, we propose the KDKE algorithm. This algorithm dynamically selects varying numbers of different modules across different layers for each distinct input sample, appends LoRA modules to the selected ones, and performs fine-tuning under multiple constraints to achieve a balance among reliability, generalization, and locality.

**Dynamic Module Selection Mechanism:** Our strategy dynamically identifies critical modules by first computing an integrated contribution score $C_{[t]}(z)$ for each module. These scores are then sorted in descending order to form a sequence $S$. The key step involves computing the gaps between adjacent scores in $S$, denoted as $\Delta S = [\delta_0, \delta_1, \ldots, \delta_{N-2}]$ where $\delta_i = s_i - s_{i+1}$. A dual-threshold criterion is applied to identify the most significant drop: a statistical threshold $\theta_{\text{gap}} = \max(0, \mu_{\Delta S} + \alpha_{\text{gap}} \cdot \sigma_{\Delta S})$ and a relative ratio threshold $\theta_{\text{ratio}} = \log(1 + \tau_{\text{ratio}})$, where $\alpha_{\text{gap}}$ and $\tau_{textratio}$ are hyperparameters. The set of critical drop points $\mathcal{P}_{\text{cfs}}$ is determined by indices where either $\delta_i > \theta_{\text{gap}}$ or $\log(\delta_i) > \theta_{\text{ratio}}$. The selected modules $\mathcal{M}_{\text{selected}}$ are then taken as all modules from the start of the sorted list up to the index of the first critical drop point (i.e., $\min(\mathcal{P}_{\text{cfs}})$).

**Constrained Adaptive Editing Algorithm:** The core idea of this algorithm is to harmonize the triad of "knowledge injection effectiveness," "preservation of original knowledge" and "generalization to new queries" through an adaptive mechanism directly aligned with the ultimate editing goal. The procedure of the algorithm is as follows: First, for a given edit request $(x_v, x_t, o^*)$ (i.e., the

---

**Algorithm 1** Knowledge-Driven Dynamic Knowledge Editing (KDKE)

**Edit Request**: $(x_v, x_t, o^*)$

**Integrated Module Contribution Score Calculation:**

$C_{[t]}(z) = \left[ C_{[t]}^P(z) \right]^m \cdot \left[ \eta \cdot C_{[t]}^O(z) + (1 - \eta) \cdot C_{[t]}^S(z) \right]^{1-m}$ for $z \in \{a_i, m_i | i = 0, ..., L\}$

**Dynamic Module Selection:**

$S \leftarrow$ Sorted $\left[ C_{[t]}(a_0), ..., C_{[t]}(a_L), C_{[t]}(m_0), ... C_{[t]}(m_L) \right]$

$\Delta S \leftarrow [\delta_0, \delta_1, \ldots, \delta_{N-2}]$, where $\delta_i = s_i - s_{i+1}$ for $i = 0, \ldots, N-2$.

$\theta_{gap} \leftarrow \max(0, \text{mean}(\Delta S) + \alpha_{gap} \cdot \text{std}(\Delta S)), \theta_{ratio} \leftarrow \log(1 + \tau_{ratio})$

$\mathcal{P}_{cfs} \leftarrow \{i \mid \delta_i \in \Delta S, \delta_i > \theta_{gap}\} \cup \{j \mid \delta_i \in \Delta S, \log(\delta_i) > \theta_{ratio}\}$

$\mathcal{M}_{selected} \leftarrow [id_0, \ldots, id_{\min(\mathcal{P}_{cfs})}]$

**Constrained Adaptive Editing:**

Retrieved similar data: $D_{sim}$

Define $f_{\theta'}$, where for $W_k \in \mathcal{M}_{selected}, W_k^e = W_k + \alpha B_k A_k$ ($A_k$ fixed, $B_k$ trainable).

**Optimize:**

$B_k^* = \arg\min_{B_k} \left\{ \frac{1}{P} \sum_{j=1}^P - \log \mathbb{P}_{f_{\theta'}} [o^* \mid p_j \oplus (x_v, x_t)] + \lambda \mathbb{E}_{x \sim D_{sim}} [D_{KL} (f_{\theta'}(x) \| f_\theta(x))] \right\}$

$f_{\theta_e} \leftarrow f_\theta$ merging $\Delta W_k^* = \alpha A_k B_k^*$ into $W_k$ for $k \in \mathcal{M}_{selected}$.

**return** $f_{\theta_e}$

---

model should output the target answer $o^*$ given inputs $x_v$ and $x_t$), the algorithm adaptively identifies a set of key intervention modules $M$ based on a dynamic module selection strategy. Then, for each target module $W \in M$, a low-rank adapter is introduced, whose weight update is defined as $\Delta W = \alpha \cdot B \cdot A$. Here, $A \in \mathbb{R}^{r \times d_{int}}$ is strictly frozen after random initialization, and $B \in \mathbb{R}^{d_{int} \times r}$ is the only trainable parameter. Previous work(Zhang et al., 2023; Zhu et al., 2024) has observed that such a design generally achieves performance comparable to training both $A$ and $B$ jointly, while significantly reducing model plasticity, constraining the optimization within a highly structured low-dimensional manifold, and fundamentally enhancing training stability.

To improve the model's generalization to diverse queries, our study constructs a prefix-augmented training distribution $\mathcal{D}_{aug}$: a set of random prefixes $\{p_j\}$ is sampled from a pre-trained language model to form augmented samples $\{(p_j \oplus x, y)\}$, thereby simulating input variations in real-world scenarios. The optimization objective of the algorithm is a multi-constraint loss function formulated as:

$$\mathcal{L} = \underbrace{\frac{1}{P} \sum_{j=1}^P - \log \mathbb{P}_{f_{\theta'}} [o^* \mid p_j \oplus (x_v, x_t)]}_{\text{Editing Loss}} + \underbrace{\lambda \cdot \mathbb{E}_{\tilde{x} \sim \mathcal{D}_{sim}} \left[ f_{\theta'}(x) \log \frac{f_{\theta'}(x)}{f_\theta(x)} \right]}_{\text{Knowledge Preservation}} \qquad (16)$$

where $\mathcal{D}_{sim}$ is a set of similar samples obtained via semantic retrieval that are irrelevant to the current edit, and the KL-divergence term ensures that the model maximally preserves the output characteristics of the original distribution when updating knowledge, thereby preventing "catastrophic forgetting".

## 4 EXPERIMENT

### 4.1 EXPERIMENTAL SETUP

**Datasets:** We conduct experiments on three multimodal knowledge editing datasets: E-IC, E-VQA(Cheng et al., 2024), and VLKEB(Huang et al., 2024). E-IC and E-VQA are constructed from multiple VQA and image captioning datasets, together with generated image data, and cover both image captioning and visual question answering tasks for targeted knowledge edits.VLKEB, in turn, provides a broader collection of vision–language editing instances with diverse entities and relations, and is specifically designed to evaluate the reliability, generalization, and locality of knowledge editing in MLLMs.

Table 1: Comparison of KDKE with existing method on the single knowledge editing task. "Rel.", "T/I-Gen." and "T/I-Loc." stand for reliability, text/modal generality, and text/modal locality, respectively. All metrics are introduced in 2.2. The t-tests demonstrate our improvements are statistically significant with $p < 0.05$ level. Additional experimental results on Qwen2.5-VL and the VLKEB benchmark across all MLLM backbones are reported in Appendix D.1.

| Method | E-IC | | | | | | E-VQA | | | | | |
|---|---|---|---|---|---|---|---|---|---|---|---|---|
| | Rel.↑ | T-Gen.↑ | I-Gen.↑ | T-Loc.↑ | I-Loc.↑ | Average | Rel.↑ | T-Gen.↑ | I-Gen.↑ | T-Loc.↑ | I-Loc.↑ | Average |
| BLIP2-OPT(2.7B) | | | | | | | | | | | | |
| FT-L | 98.63 | 94.31 | **99.89** | 43.40 | 19.65 | 71.18 (±1.23) | 100 | **99.92** | 98.97 | 67.92 | 36.76 | 78.71 (±1.17) |
| FT-V | 34.26 | 60.62 | 34.07 | 100 | 40.45 | 53.88 (±1.42) | 47.26 | 36.74 | 46.99 | 100 | 30.20 | 52.24 (±1.35) |
| KE | 69.02 | 62.80 | 61.22 | 96.21 | 45.55 | 66.96 (±0.87) | 67.81 | 63.00 | 66.17 | 97.32 | 45.89 | 68.04 (±0.92) |
| IKE | 96.70 | 78.20 | 83.15 | 13.36 | 2.17 | 54.72 (±1.38) | 99.95 | 91.59 | 92.33 | 13.16 | 1.88 | 59.78 (±1.26) |
| WilKE | 64.03 | 30.76 | 63.98 | 93.00 | 61.67 | 62.69 (±0.95) | 66.45 | 28.91 | 62.34 | 91.25 | 59.87 | 61.76 (±0.89) |
| SERAC | 94.40 | 95.98 | 91.49 | 100 | 0.47 | 76.58 (±1.05) | 91.22 | 91.40 | 89.81 | 100 | 0.33 | 74.55 (±1.12) |
| MEND | 65.13 | 38.00 | 36.19 | 92.67 | 55.72 | 57.54 (±1.28) | 92.60 | 90.80 | 91.94 | 96.07 | 65.15 | 87.31 (±0.76) |
| TP | 49.71 | 49.03 | 45.46 | 93.88 | 80.88 | 63.79 (±0.83) | 68.31 | 60.88 | 56.35 | 98.49 | 85.27 | 73.86 (±0.79) |
| LTE | 96.69 | 95.26 | 94.06 | 95.25 | 87.68 | 93.79 (±0.52) | 97.74 | 97.21 | 96.35 | 94.34 | 84.99 | 94.13 (±0.48) |
| VisEdit | 97.06 | **96.88** | 94.85 | **100** | 91.74 | 96.11 (±0.41) | 98.01 | 97.57 | 94.71 | **100** | 91.26 | 96.31 (±0.45) |
| **KDKE** | **100** | 94.06 | 98.55 | 96.25 | 95.17 | **96.81** (±0.36) | **100** | 94.33 | **99.04** | 96.32 | 92.97 | **97.13** (±0.39) |
| LLaVA-v1.5(7B) | | | | | | | | | | | | |
| FT-L | 100 | 82.15 | 81.12 | 13.36 | 2.17 | 55.76 (±0.83) | 100 | 88.23 | 88.39 | 76.34 | 23.48 | 75.29 (±1.27) |
| FT-V | 47.26 | 36.74 | 46.99 | 100 | 30.18 | 52.24 (±1.42) | 50.75 | 50.07 | 48.66 | 100 | 1.09 | 50.11 (±0.67) |
| KE | 83.54 | 82.15 | 81.12 | 13.36 | 2.17 | 52.47 (±1.11) | 85.86 | 84.05 | 82.23 | 93.57 | 73.06 | 83.75 (±0.94) |
| IKE | 93.72 | 88.37 | 76.99 | 76.58 | 64.90 | 79.92 (±0.59) | 91.35 | 90.84 | 91.08 | 60.18 | 51.08 | 76.91 (±1.33) |
| WilKE | 63.80 | 65.48 | 61.33 | 87.30 | 88.83 | 73.35 (±1.08) | 64.72 | 67.83 | 60.45 | 86.91 | 77.49 | 71.48 (±0.72) |
| SERAC | 43.08 | 42.37 | 42.85 | 100 | 7.63 | 47.19 (±1.46) | 82.51 | 81.63 | 80.05 | 100 | 57.48 | 78.33 (±1.46) |
| MEND | 93.76 | 93.46 | 92.14 | 91.63 | 87.59 | 91.71 (±1.24) | 92.33 | 92.16 | 92.13 | 90.29 | 81.13 | 89.61 (±0.53) |
| TP | 59.07 | 57.01 | 55.51 | 64.79 | 89.26 | 65.13 (±0.48) | 38.68 | 36.27 | 31.26 | 95.31 | 91.41 | 58.59 (±1.19) |
| LTE | 93.60 | 92.38 | 91.18 | 85.54 | 88.49 | 90.24 (±1.37) | 94.16 | 93.54 | 93.06 | 83.76 | 81.65 | 89.23 (±0.85) |
| VisEdit | 95.27 | **94.64** | 93.57 | 100 | 96.10 | 95.92 (±0.76) | 95.99 | 95.78 | **94.71** | 100 | 94.12 | 94.12 (±1.03) |
| **KDKE** | **100** | 93.54 | **96.87** | 97.45 | 96.59 | **96.89** (±1.41) | **100** | 96.17 | 92.56 | 96.69 | 95.87 | **96.26** (±0.62) |
| MiniGPT-4(7B) | | | | | | | | | | | | |
| FT-L | 100 | 92.64 | 90.56 | 29.34 | 11.65 | 64.84 (±1.23) | 100 | 95.21 | 93.84 | 30.95 | 25.04 | 69.01 (±0.87) |
| FT-V | 46.69 | 45.58 | 44.02 | 100 | 90.85 | 65.43 (±0.95) | 27.12 | 22.04 | 21.75 | 100 | 87.80 | 51.74 (±1.42) |
| KE | 35.10 | 24.20 | 5.89 | 96.78 | 52.22 | 42.84 (±1.38) | 87.77 | 86.62 | 3.76 | 97.15 | 55.77 | 66.21 (±1.15) |
| IKE | 68.60 | 59.80 | 63.58 | 12.51 | 2.96 | 41.49 (±0.68) | 71.72 | 40.23 | 70.59 | 13.46 | 2.00 | 39.60 (±1.27) |
| WilKE | 72.18 | 31.67 | 52.89 | 78.34 | 81.56 | 63.33 (±1.09) | 68.33 | 45.89 | 47.65 | 75.49 | 77.88 | 63.05 (±0.73) |
| SERAC | 40.20 | 36.60 | 35.12 | 100 | 0.97 | 42.58 (±1.46) | 87.20 | 84.60 | 81.87 | 100 | 0.33 | 70.80 (±0.59) |
| MEND | 87.10 | 84.10 | 80.60 | 98.34 | 59.53 | 81.93 (±0.52) | 95.51 | 95.27 | 86.37 | 98.73 | 71.33 | 79.44 (±1.34) |
| TP | 52.55 | 51.93 | 49.65 | 85.56 | 72.66 | 62.47 (±1.17) | 42.96 | 41.53 | 40.70 | 93.32 | 84.61 | 60.62 (±0.81) |
| LTE | 89.68 | 87.48 | 86.15 | 86.52 | 87.48 | 87.46 (±0.45) | 95.92 | 95.25 | 94.92 | 87.18 | 89.72 | 92.60 (±1.28) |
| VisEdit | 93.25 | 90.32 | 89.75 | **100** | 94.09 | 93.48 (±1.03) | 96.83 | 96.46 | 95.38 | **100** | 90.88 | 95.91 (±0.64) |
| **KDKE** | **100** | **94.30** | **93.03** | 96.09 | 92.27 | **95.14** (±0.76) | **100** | **97.12** | **97.76** | 95.07 | 91.16 | **96.22** (±1.11) |

**MLLM Backbones:** To ensure a comprehensive evaluation and align with existing works, we select MLLM backbones of varying architectures and parameter scales, including BLIP2-OPT (2.7B)(Li et al., 2023), LLaVA-V1.5 (7B)(Liu et al., 2023), MiniGPT-4 (7B)(Zhu et al., 2023) and Qwen2.5-VL(3B)(Bai et al., 2025).

**Baseline Editors:** VisEdit(Chen et al., 2025b) is an editing methods specifically designed for MLLMs, we also introduced adapted LLM editors to compare, including FT-V, FT-L, KE(Cao et al., 2021), IKE(Zheng et al., 2023), WilKE(Hu et al., 2024), SERAC(Mitchell et al., 2022b), MEND(Mitchell et al., 2022a), TP(Huang et al., 2023) and LTE(Jiang et al., 2024).

## 4.2 ANALYSIS OF EDITING PERFORMANCE

The overall editing performance is exhibited in Table 1, Table 5 and Table 6. Here we analyze Table 1 from different perspectives.

From the perspective of editors, a fundamental challenge in knowledge editing has been the difficulty in simultaneously optimizing the three key metrics—reliability (Rel.), generality (Gen.), and locality (Loc.)—resembling an "impossible triangle". Multimodal knowledge editing further expands the evaluation to five metrics, increasing the complexity of the challenge. Previous methods that focus solely on reliability and generality, such as FT-L, which fully fine-tunes the last layer of the model, achieve nearly perfect performance on these two metrics at the cost of severe overfitting. In contrast, IKE prepends the editing knowledge and its corresponding answer as a prompt prefix for single-instance editing, but its performance on out-of-scope samples is constrained by this prefix, leading to a low locality score. Moreover, since FT-V and VisEdit only modify the visual components, and SERAC uses a classifier to distinguish pure text inputs, they effectively avoid interference

Table 2: Ablation results of of BLIP2-OPT on E-VQA and E-IC datasets.

| Settings | E-IC | | | | | E-VQA | | | | |
|---|---|---|---|---|---|---|---|---|---|---|
| | Rel.↑ | T-Gen.↑ | I-Gen.↑ | T-Loc.↑ | I-Loc.↑ | Rel.↑ | T-Gen.↑ | I-Gen.↑ | T-Loc.↑ | I-Loc.↑ |
| **KDKE** | **100** | **94.06** | 98.55 | **96.25** | **95.17** | **100** | 94.33 | 99.04 | **96.32** | **92.97** |
| latest mlp | 96.47 | 87.33 | 96.08 | 94.54 | 81.92 | 99.67 | 92.12 | 94.75 | 89.42 | 74.02 |
| w/o KL | 100 | 93.67 | 99.10 | 87.26 | 60.88 | 100 | **99.00** | **99.67** | 72.99 | 40.08 |
| finetune | 100 | 94.06 | **100** | 65.15 | 43.37 | 92.07 | 91.90 | 92.09 | 47.98 | 13.37 |
| mlp only | 100 | 93.22 | 99.72 | 93.00 | 77.89 | 100 | 93.56 | 99.03 | 96.26 | 51.17 |
| att only | 100 | 92.88 | 98.76 | 80.22 | 33.12 | 100 | 92.77 | 96.62 | 89.13 | 30.81 |

with textual locality samples. Although our editor does not achieve the best performance on every single metric, it attains the highest average performance across all metrics while maintaining high reliability, thanks to our proposed dynamic module selection algorithm.

From the perspective of backbones, the editing effectiveness varies slightly across BLIP2-OPT, LLaVA-V1.5, and MiniGPT-4 due to differences in model size and architecture. Taking the E-VQA dataset as an example, image generalization capability is higher in BLIP2-OPT and MiniGPT-4, while it is significantly lower in LLaVA. This can be attributed to LLaVA's use of CLIP-ViT as the visual encoder, which, unlike those used in BLIP2 and MiniGPT-4, establishes a tighter and more fixed mapping between visual features and the language space. Consequently, when editing the language model part, the adaptability to visual features is relatively weaker, affecting its generalization ability to unseen image combinations.

From the perspective of datasets, performance in image locality also varies across different datasets. Specifically, image locality is notably higher in E-IC than in E-VQA. This is because the QA pairs in E-VQA involve broader and more complex visual concepts and semantic relationships. The fine-tuning-based editing process may cause slight perturbations to the model's original ability to handle these complex associations. In contrast, the knowledge points in E-IC are more isolated and focused. Fine-tuning edits on such data have a smaller impact on other unrelated parameters, thus better preserving the model's original performance on non-edited data. This observation aligns with our analysis based on the multi-level module contribution scores for both datasets.

### 4.3 ABLATION STUDY

To validate the effectiveness of the two core components of our framework—the Dynamic Module Selection mechanism and the Constrained Adaptive Editing algorithm. We conduct a series of ablation experiments on BLIP2-OPT using both E-IC and E-VQA datasets. The results are presented in Table 2.

**Ablation on Constrained Adaptive Editing:** We first investigated the contribution of the constrained adaptive editing algorithm. Replacing LoRA with direct fine-tuning significantly declined locality metrics, indicating severe overfitting. Similarly, removing the KL-divergence constraint ("w/o KL") also degraded locality , despite marginally improved generalization. These results confirm that both low-rank parameterization and the distribution-matching constraint are essential for balancing editing efficacy and knowledge preservation.

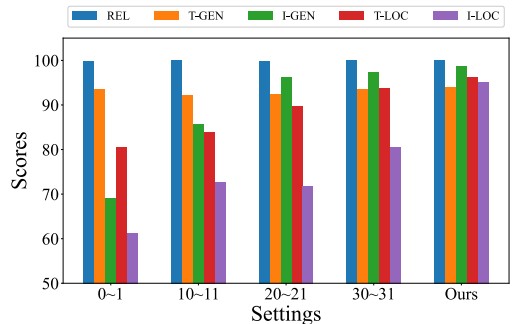

Figure 3: Ablation of Layer Select using E-IC on BLIP2-OPT with sliding window length 2.

**Ablation on Dynamic Module Selection:** We then evaluated the impact of the dynamic module selection strategy. Using static selection strategies—such as editing only the last MLP layer ("latest mlp"), only MLP layers ("mlp only"), or only attention layers ("att only")—consistently resulted in inferior performance across all metrics compared to our full method. For instance, on E-VQA, "att

only" caused I-Loc. to drop to 30.81. This demonstrates that adaptively selecting both attention and MLP modules across different layers is critical for effective and precise knowledge updates.

**Layer-wise Selection Analysis:** We further analyzed the effect of selecting modules from different parts of the network. Using a fixed-length sliding window (e.g., 2 consecutive layers, as shown in Figure 3 ) in the early, middle, or late layers of the model consistently underperformed compared to our dynamic selection approach. This confirms that the importance of modules is instance-specific, and a fixed set of layers cannot achieve optimal results. More analysis is in Appendix D.

In summary, the ablations confirm that both the dynamic selection mechanism and the constrained editing algorithm are indispensable components of KDKE, each contributing significantly to the overall editing performance.

## 4.4 SENSITIVITY ANALYSIS

To evaluate the robustness of *KDKE* to hyperparameter variations, we conduct a sensitivity analysis on key parameters in the Dynamic Module Selection mechanism: the gap threshold coefficient $\alpha_{\text{gap}}$ (controlling statistical drop detection) and the ratio threshold $\tau_{\text{ratio}}$ (controlling relative drop detection). We vary $\alpha_{\text{gap}}$ in $\{0.5, 1.0, 1.5\}$ and $\tau_{\text{ratio}}$ in $\{0.1, 0.5, 1.0\}$, while fixing other hyperparameters to their default values ($\alpha_{\text{gap}} = 1.0$, $\tau_{\text{ratio}} = 0.5$). Experiments are performed on BLIP2-OPT using the E-IC dataset, reporting the average metric score (as defined in Section 2.2).

Table 3: Sensitivity analysis of KDKE to hyperparameter variations on BLIP2-OPT (E-IC).

| **Parameter** | $\alpha_{\text{gap}} = 0.5$ | $\alpha_{\text{gap}} = 1.0$ | $\alpha_{\text{gap}} = 1.5$ |
|---|---|---|---|
| $\tau_{\text{ratio}} = 0.1$ | 95.23 ($\pm$0.45) | 96.12 ($\pm$0.38) | 94.87 ($\pm$0.52) |
| $\tau_{\text{ratio}} = 0.5$ | 96.45 ($\pm$0.39) | **96.81** ($\pm$0.36) | 95.98 ($\pm$0.41) |
| $\tau_{\text{ratio}} = 1.0$ | 95.67 ($\pm$0.47) | 96.34 ($\pm$0.40) | 95.12 ($\pm$0.48) |

As shown in Table 3, performance remains stable across the tested ranges, with variations under $1.5\%$ in average score. The default settings yield the highest performance, but even at extremes (e.g., $\alpha_{\text{gap}} = 1.5$, $\tau_{\text{ratio}} = 0.1$), the method retains over $94\%$ of optimal efficacy. This demonstrates *KDKE*'s robustness to threshold perturbations, ensuring reliable deployment in diverse scenarios without extensive hyperparameter tuning. Additional sensitivities (e.g., to $\eta$ and $m$ in the contribution score) are analyzed in Appendix D, confirming consistent trends.

## 5 RELATED WORKS

### 5.1 MULTIMODAL LARGE LANGUAGE MODELS

Multimodal Large Language Models (MLLMs) integrate visual encoders with Large Language Models (LLMs) to process and generate cross-modal content. Early models like CLIP (Radford et al., 2021) established a shared image-text semantic space through contrastive learning. Subsequent architectures have developed more efficient fusion strategies: BLIP-2 (Li et al., 2023) introduced a query-based transformer (Q-Former) as an information bottleneck to efficiently project visual features into a frozen LLM. LLaVA-v1.5 (Liu et al., 2023) adopted a minimalist "early fusion" approach using a linear projection layer, demonstrating that high-quality instruction tuning can compensate for architectural simplicity. In contrast, MiniGPT-4 (Zhu et al., 2023) employs a "late fusion" strategy with multi-stage training to foster open-ended textual generation from visual inputs, highlighting a trade-off between output richness and training complexity.Building on these advancements, Qwen2.5-VL (Bai et al., 2025) leverages a scalable vision-language architecture with native support for high-resolution images and multilingual capabilities, achieving superior performance in complex visual reasoning tasks through unified pre-training and efficient parameter scaling.

### 5.2 KNOWLEDGE EDITING IN LARGE LANGUAGE MODELS

Knowledge editing seeks to update model behavior without full retraining. Methods can be categorized as follows: **Direct parameter modification** locates and alters specific parameters. ROME (Meng et al., 2023a) uses causal tracing to apply low-rank updates to MLP layers; MEMIT (Meng

et al., 2023b) extends this to mass edits. To minimize unintended side-effects, AlphaEdit (Fang et al., 2025) formulates editing as constrained optimization. **Meta-learning** approaches, like MEND, learn to predict parameter updates via a hypernetwork. **Additional parameters** methods (e.g., LEMOE (Wang & Li, 2024)) introduce new trainable components to avoid altering original weights. **External modules** store edits separately and integrate them during inference, as in SERAC (Mitchell et al., 2022b) and GRACE(Hartvigsen et al., 2023). **In-context editing** techniques (e.g., ICE(Qi et al., 2025), MeLLo(Zhong et al., 2024)) prepend edits to the input prompt, leveraging in-context learning for temporary updates.

**Knowledge Editing in MLLMs** is an emerging area. MMEdit (Cheng et al., 2024) and VLKEB (Huang et al., 2024) contributed evaluation benchmarks. VisEdit (Chen et al., 2025b) edits intermediate visual representations, while LiveEdit (Chen et al., 2025a) uses a Mixture-of-Experts structure for lifelong updating. However, efficient direct parameter editing methods tailored for MLLMs remain underexplored, a gap our work addresses.

## 6 CONCLUSION

In this work, we tackled the fundamental challenges of knowledge editing in Multimodal Large Language Models (MLLMs): the distributed nature of cross-modal knowledge makes it difficult to localize, and uncontrolled editing often compromises generalization and locality. We introduced KDKE, a dynamic editing framework that moves beyond static, one-size-fits-all updates. At its core, KDKE uses a novel Integrated Module Contribution Score to accurately attribute knowledge to specific modules and a Dynamic Module Selection mechanism to adaptively identify critical parameters per instance. This precise localization is coupled with the Constrained Adaptive Editing algorithm, localized fine-tuning via injected LoRA parameters regularized by a multi-objective loss, ensuring high reliability, generalization, and minimal side effects.

Our extensive evaluations on multiple architecturesand benchmarks confirm that KDKE achieves state-of-the-art performance, effectively balancing the critical trade-offs in knowledge editing. This work establishes a new paradigm for efficient, precise, and safe knowledge updates in MLLMs, paving the way for more adaptable and maintainable multimodal systems.

## 7 ETHICS STATEMENT

We confirm that this work complies with the ICLR Code of Ethics and, to the best of our knowledge, does not raise any foreseeable ethical concerns.

## 8 REPRODUCIBILITY STATEMENT

To guarantee reproducibility, we have made the following efforts. (1) The pseudo-code for our method is provided in 1. (2) All datasets used in this work are publicly available, and we provide a detailed description in 4.1 (3) We commit to releasing the full source code of our proposed algorithm and training framework upon acceptance of the paper.

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

## A LLM USAGE

In the preparation of this work, the authors used LLMs primarily as a tool for polishing and refining the English language expression. The LLM was employed to assist in rephrasing certain sentences to improve clarity, fluency, and academic tone. It is important to note that all core ideas, theoretical developments, experimental design, results analysis, and conclusions are solely those of the authors. The LLM did not contribute to the intellectual substance of the research, nor was it used for generating any scientific content or ideation. The final manuscript has been thoroughly reviewed and approved by all authors.

## B RELATED PROOF, ANALYSIS AND DISCUSSION

This appendix aims to provide a detailed theoretical foundation, mathematical derivation, and experimental analysis supporting the effectiveness and interpretability of the "Integrated Module Contribution Score".

### B.1 DETAILED THEORETICAL DERIVATION OF THE INTEGRATED MODULE CONTRIBUTION SCORE

This section will elaborate on the mathematical definitions, derivation processes, and design motivations for each component of the integrated module contribution score, ensuring the rigor of its theoretical basis.

#### B.1.1 TRANSFORMER RESIDUAL STREAM AND LOGIT DECOMPOSITION

A core principle of the Transformer architecture is its residual connections, which allow information to flow efficiently between layers and mitigate the vanishing gradient problem. In a Transformer, the hidden state of each token is updated through a residual stream from layer to layer. Specifically, given an input sequence, after token embedding and positional encoding, it forms the initial hidden state $h^0$. In each Transformer layer $l$, both the Attention and MLP modules read from and write incremental updates to the residual stream.

For the last token $[N]$ in the sequence (typically used for next token prediction), its hidden state $h_{[N]}^l$ in layer $l$ can be expressed as a residual connection of the previous layer's state plus the output of the current layer's Attention and MLP modules. However, a deeper observation reveals that the final layer's hidden state $h_{[N]}^L$ can be viewed as the sum of the initial hidden state $h_{[N]}^0$ and the accumulated outputs of all intermediate Attention modules $a_{[N]}^l$ and MLP modules $m_{[N]}^l$ at their respective layers:

$$h_{[N]}^L = h_{[N]}^0 + \sum_{l=1}^{L} a_{[N]}^l + \sum_{l=1}^{L} m_{[N]}^l \tag{17}$$

This decomposition is fundamental to understanding module contributions. In Transformer models, the final token prediction distribution is obtained by projecting $h_{[N]}^L$ into a vocabulary-sized vector (logits) and applying a softmax operation. This projection is typically performed by a linear layer

$W_V$. By multiplying both sides of Equation equation 17 by $W_V$, we can decompose the final logit vector $h_{[N]}^L W_V$:

$$h_{[N]}^L W_V = h_{[N]}^0 W_V + \sum_{l=1}^{L} a_{[N]}^l W_V + \sum_{l=1}^{L} m_{[N]}^l W_V \tag{18}$$

Equation equation 18 clearly shows that the final predicted logit values can be decomposed into the contribution of the initial input $h_{[N]}^0$ to the logits, and the incremental contributions of each Attention module $a_{[N]}^l$ and MLP module $m_{[N]}^l$ within each layer. This provides a direct mathematical basis for quantifying each module's contribution to a specific target token. Each $(a_{[N]}^l W_V)$ and $(m_{[N]}^l W_V)$ term can be considered the "raw contribution" tensor of that module to the final logit vector.

### B.1.2 DERIVATION AND DISCUSSION OF MAPPING PROBABILITY

As the first measure of module contribution, we define the mapping probability $C_{[t]}^P(z)$, which directly quantifies the predicted probability of the target token $t$ after the module's output $z$ is projected by $W_V$ and subjected to a softmax operation.

$$C_{[t]}^P(z) = \text{softmax}(zW_V)_{[t]} \tag{19}$$

Here, $z$ represents the output of a module (e.g., $a_{[N]}^l$ or $m_{[N]}^l$), and $(zW_V)_{[t]}$ is the logit value corresponding to the target token $t$ after the module's output is projected by $W_V$.

**Discussion:** $C_{[t]}^P(z)$ is the most intuitive metric for module contribution, as it answers the question: "If only considering this module's output, what is the likelihood that the model predicts the target token?" Its advantage lies in its direct correlation with the model's ultimate prediction behavior (probability distribution), making it easy to understand. From a **probability viewpoint**, it reflects the module's direct influence at the output layer. However, it has a crucial limitation: when all logit values produced by the module are very small (e.g., all negative and large in magnitude), even a relatively low logit value for the target token $t$ (i.e., $(zW_V)_{[t]}$) can result in a comparatively high probability after softmax normalization. This can lead to misjudging the module's actual contribution; it might "appear" to contribute significantly, but its absolute impact on the final logits and model decision could be negligible. To address this issue, we introduce additional contribution components.

### B.1.3 DERIVATION AND DISCUSSION OF NORMALIZED LOGIT VALUE

To overcome the limitations of $C_{[t]}^P(z)$, we introduce the normalized logit value $C_{[t]}^O(z)$, aimed at measuring the absolute strength of a module's logit contribution to the target token $t$, rather than just its relative probability.

$$C_{[t]}^O(z) = \frac{(zW_V)_{[t]}}{\max_{l=1}^L \left( |(a_{[N]}^l W_V)_{[t]}|, |(m_{[N]}^l W_V)_{[t]}| \right)} \tag{20}$$

Here, the numerator $(zW_V)_{[t]}$ directly represents the raw logit contribution of module $z$ to the target token $t$. The denominator is a global normalization factor, computed as the maximum absolute logit contribution to the target token $t$ across all Attention and MLP modules in all layers.

**Discussion:** The design of $C_{[t]}^O(z)$ is to assess the "raw strength" of a module's contribution, quantifying the module's direct gain in the logit space from a **logit viewpoint**. By normalizing a module's contribution against the strongest absolute logit contribution to target token $t$ observed across all modules, this metric effectively scales the contribution strength to a comparable range. If a module's $(zW_V)_{[t]}$ value is high, even if its $C_{[t]}^P(z)$ appears unremarkable due to other competing tokens also having high logits, $C_{[t]}^O(z)$ can still identify its strong absolute support. Conversely, if a module has a high $C_{[t]}^P(z)$ but a low $C_{[t]}^O(z)$, it indicates that its high probability arises because other tokens' logits are even lower, rather than from providing strong positive support itself. Therefore, a high $C_{[t]}^O(z)$ implies that the module provides substantial absolute support for the target token.

### B.1.4 Derivation and Discussion of Significant Contribution

Beyond absolute logit strength, a module's "definitive support" for the target token is also crucial. A module might assign high probability and high absolute logit to the target token, but if it also assigns similarly high scores to many other tokens, its unique support for the target token is not sufficiently strong. To capture this "discriminative power," we introduce the significant contribution $C_{[t]}^S(z)$.

First, we define the Logit Gap $D_{[t]}(z)$:

$$D_{[t]}(z) = (zW_V)_{[t]} - \max_{k \in \mathcal{V} \setminus \{[t]\}} (zW_V)_k \tag{21}$$

This gap measures the difference between the module $z$'s logit for the target token $t$ and its logit for the second-highest scoring token in the vocabulary. A large $D_{[t]}(z)$ indicates that the module clearly prefers the target token $t$.

Next, we define the significant contribution $C_{[t]}^S(z)$ by normalizing this logit difference and ensuring non-negativity:

$$C_{[t]}^S(z) = \frac{\text{ReLU}(D_{[t]}(z))}{\max_{l=1}^{L} \max\left(\text{ReLU}(D_{[t]}(a_{[N]}^l)), \text{ReLU}(D_{[t]}(m_{[N]}^l))\right) + \epsilon} \tag{22}$$

Here, $\text{ReLU}(D_{[t]}(z))$ ensures that a positive significant contribution is only counted if the logit of the target token $t$ is indeed the highest (i.e., $D_{[t]}(z) > 0$). If $D_{[t]}(z) \le 0$, it implies that the module does not unequivocally rank the target token first, and thus its significant contribution is zero. The denominator is again a global normalization factor, representing the maximum positive Logit Gap for the target token $t$ across all modules. A small constant $\epsilon$ is added for numerical stability, preventing division by zero.

**Discussion:** $C_{[t]}^S(z)$ effectively quantifies a module's "discriminative power" or "uniqueness of support" for the target token, also reflecting the module's decisiveness from a **logit viewpoint**. A high $C_{[t]}^S(z)$ suggests that the module not only supports the target token but also clearly distinguishes it from all other possible tokens, providing "definitive support." This metric addresses the distinction between scenarios where a module offers moderate support for the target token but very weak support for all others, versus scenarios where it provides strong support for the target token but also strong support for several other tokens. The latter, while potentially generating high probability, lacks decisiveness.

### B.1.5 Combination Mechanism and Parameter Discussion for the Integrated Contribution Score

To provide a comprehensive and robust evaluation of module contributions, we combine the three aforementioned components into a final integrated module contribution score $C_{[t]}(z)$:

$$C_{[t]}(z) = \left[C_{[t]}^P(z)\right]^m \cdot \left[\eta \cdot C_{[t]}^O(z) + (1 - \eta) \cdot C_{[t]}^S(z)\right]^{1-m} \tag{23}$$

**Combination Mechanism:** The combination mechanism of this method strategically merges contributions from different "viewpoints." It first performs a **weighted arithmetic mean** on components derived from the same viewpoint, and then applies a **weighted geometric mean** to combine scores from different viewpoints.

**Weighted Arithmetic Mean for Logit-Viewpoint Components:** We first combine $C_{[t]}^O(z)$ (normalized logit value, reflecting absolute logit strength) and $C_{[t]}^S(z)$ (significant contribution, reflecting logit discriminative power) through a weighted arithmetic mean: $\eta \cdot C_{[t]}^O(z) + (1 - \eta) \cdot C_{[t]}^S(z)$. The choice of an **arithmetic mean** is appropriate here because both $C_{[t]}^O(z)$ and $C_{[t]}^S(z)$ originate from the **logit viewpoint**. They measure different but complementary aspects of a module's support for the target token within the logit space. $C_{[t]}^O(z)$ focuses on the module's direct "force" in terms of logit values, while $C_{[t]}^S(z)$ addresses the "clarity" or "exclusivity" of the module's logit-based decision. These are factors that can be considered additive or co-acting at the logit level. Using

an arithmetic mean allows a module to excel in one aspect, thereby boosting its overall logit-level support, and its comprehensive contribution can be adjusted by weights even if another aspect is weaker.

**Weighted Geometric Mean for Different Viewpoint Components:** Subsequently, we combine $C_{[t]}^P(z)$ (from the **probability viewpoint**) with the arithmetically averaged deep support term (from the **logit viewpoint**) using a weighted geometric mean. The choice of a **geometric mean** (multiplication) is more suitable for combining measures that originate from different natures or scales, especially when all these measures must reach a certain threshold to constitute a "comprehensive" contribution.

- **Complementarity Requirement:** $C_{[t]}^P(z)$ assesses the module's "surface" predictive performance, i.e., its behavior at the final output probability level. In contrast, the deep support term reflects the module's "deeper," more intrinsic logit-level support. A truly contributory module should perform well from both viewpoints. If a module has a very high probability score (e.g., because all other tokens' logits are extremely low) but its deep logit support ($C_{[t]}^O$ and $C_{[t]}^S$) is weak, its actual contribution is questionable. The converse is also true. The multiplicative nature of the geometric mean ensures that if the score from any single viewpoint (or its combination) approaches zero, the final integrated score will also be small, effectively penalizing modules that underperform in a critical viewpoint.

- **Robustness:** This combination method offers better robustness against outliers and better reflects the "synergistic effect" of module contributions.

**Role and Impact of Parameter** $m$**:** The parameter $m \in [0, 1]$ controls the relative weight between the direct mapping probability $C_{[t]}^P(z)$ and the deep support term $[\eta \cdot C_{[t]}^O(z) + (1 - \eta) \cdot C_{[t]}^S(z)]$.

- When $m$ approaches 1, the integrated score will place a greater emphasis on the module's direct predictive capability from the **probability viewpoint**, $C_{[t]}^P(z)$.

- When $m$ approaches 0, the integrated score will prioritize the deep support term from the **logit viewpoint**.

By adjusting $m$, one can fine-tune the emphasis of the evaluation based on different priorities regarding the concept of "contribution" (e.g., whether to focus more on the module's superficial predictive capability or its deeper decision-making mechanisms).

**Role and Impact of Parameter** $\eta$**:** The parameter $\eta \in [0, 1]$ is used to balance the weights between $C_{[t]}^O(z)$ and $C_{[t]}^S(z)$, which are both **logit-viewpoint** components, within the deep support term:

- When $\eta$ approaches 1, the deep support term focuses more on the absolute strength of logits, $C_{[t]}^O(z)$.

- When $\eta$ approaches 0, the deep support term places more emphasis on the significant discriminative power of logits, $C_{[t]}^S(z)$.

This parameter allows users or researchers to decide, based on specific analytical needs, whether to prioritize the absolute logit gain provided by a module or its ability to differentiate the target token from other competing tokens.

Through this layered combination mechanism, the integrated contribution score provides a flexible and configurable framework that can comprehensively evaluate the contribution of Transformer modules to specific token predictions, overcoming the limitations of single metrics and ensuring the validity and robustness of the assessment across different analytical viewpoints.

## B.2 THEORETICAL PROPERTIES AND ASSUMPTIONS

This section will discuss the theoretical assumptions and key properties underpinning the integrated module contribution score method.

### B.2.1 Assumption of Linear Superposition in the Residual Stream

The core premise of this method is that the final hidden state of the Transformer $h_{[N]}^L$ and, consequently, the final logit vector $h_{[N]}^L W_V$, can be precisely decomposed into a linear superposition of the initial input and the contributions from all intermediate Attention and MLP modules (as shown in equation 17 and equation 18). This assumption holds true in standard Transformer architectures because residual connections are additive linear operations, and each module's output (before entering the residual stream) is computed independently through its weight matrices. This ensures that the contribution of each module $a_{[N]}^l$ or $m_{[N]}^l$ to the final logits is separable and additive. This linearity is the foundational basis for performing module-level attribution and quantification.

### B.2.2 Monotonicity of the Softmax Function

The softmax function, defined as $\text{softmax}(x)_i = \frac{e^{x_i}}{\sum_j e^{x_j}}$, exhibits monotonicity. This means that if an element $x_i$ in a vector increases, its corresponding softmax probability $\text{softmax}(x)_i$ will also increase (assuming other elements remain constant or change at a slower rate). More importantly, the softmax function preserves the relative ordering of input logits: if $x_i > x_j$, then $\text{softmax}(x)_i > \text{softmax}(x)_j$. This property guarantees that our analysis of module output logit values can indirectly reflect their impact on the final probability distribution. When a module significantly increases the logit value of a target token, it invariably boosts the prediction probability of that token. Therefore, even though $C_{[t]}^P(z)$ has the aforementioned limitations, its basic directional agreement with logit magnitudes remains consistent, providing a logical basis for introducing more refined logit-based metrics.

### B.2.3 Selection of Normalization Factors

In this method, both $C_{[t]}^O(z)$ and $C_{[t]}^S(z)$ utilize global normalization factors:

- For $C_{[t]}^O(z)$, the denominator is the maximum absolute logit contribution to the target token across all modules: $\max_{l=1}^L \left( |(a_{[N]}^l W_V)_{[t]}|, |(m_{[N]}^l W_V)_{[t]}| \right)$.

- For $C_{[t]}^S(z)$, the denominator is the maximum positive Logit Gap for the target token across all modules: $\max_{l=1}^L \max \left( \text{ReLU}(D_{[t]}(a_{[N]}^l)), \text{ReLU}(D_{[t]}(m_{[N]}^l)) \right) + \epsilon$.

The primary purpose of selecting these global normalization factors is to ensure comparability of contribution scores across different modules. By comparing each module's contribution against the maximum observed value from all modules within the entire model, we scale the scores into a normalized range (typically $[0, 1]$, or potentially $[-1, 1]$ when negative contributions are considered for $C^O$). This enables a fair comparison of the strength and uniqueness of contributions from different layers and module types for the same target token. This approach avoids misleading comparisons that might arise due to inherent differences in the output value ranges of different modules.

### B.2.4 Robustness to Small Logit Values

This method addresses the sensitivity to small logit values, which is a limitation of relying solely on $C_{[t]}^P(z)$, by introducing $C_{[t]}^O(z)$ and $C_{[t]}^S(z)$.

- $C_{[t]}^O(z)$ directly focuses on the raw logit value $(zW_V)_{[t]}$ output by the module. Through global maximum normalization, it accurately reflects the absolute contribution, even if all logits are small. It does not get "inflated" merely because other non-target tokens have even smaller logits.

- $C_{[t]}^S(z)$ further emphasizes the "decisiveness" of the module for the target token through the Logit Gap $D_{[t]}(z)$ and the ReLU operation. Only when a module truly makes the target token's logit significantly higher than other tokens will it receive a high $C_{[t]}^S(z)$. This makes the metric robust against situations where a relatively high probability arises from weak signals.

- **Integrated Viewpoint:** Collectively, the combined design of these three components, particularly the introduction of $C_{[t]}^O(z)$ and $C_{[t]}^S(z)$ and their geometric mean combination with $C_{[t]}^P(z)$, allows the integrated module contribution score to more accurately reflect a module's true influence, rather than merely superficial probability values. By demanding strong performance at both the probability level and the logit level, this scoring system effectively filters out "spurious high-probability" contributions that lack deep underlying support, thereby enhancing robustness and the accuracy of interpretation.

### B.3 VISUALIZATION OF SELECTED MODULES

Figures 4 and 5 illustrate the modules selected by our approach, which combines an integrated module contribution score with a dynamic module selection mechanism, applied to BLIP2-OPT on the E-VQA and E-IC tasks respectively. On E-VQA, the selected modules are distributed across a variety of layers, with a predominant concentration in the later layers (15–32). In contrast, for E-IC, where the text prompt is consistently fixed as "a photo of", the selected modules are more concentrated and exhibit less variation across layers.

## C HYPERPARAMETER SETTINGS

We use the following default hyperparameters for KDKE, tuned via grid search on a validation split of E-IC and E-VQA. All experiments employ AdamW optimizer with learning rate $1 \times 10^{-4}$, batch size 1, and 100 optimization steps unless specified otherwise.

Table 4: Default hyperparameters for KDKE.

| Parameter | Default Value | Range | Description |
|---|---|---|---|
| $\alpha_{\text{gap}}$ | 1.0 | [0.5, 1.5] | Gap threshold coefficient for dynamic module selection (statistical drop detection). |
| $\tau_{\text{ratio}}$ | 0.5 | [0.1, 1.0] | Ratio threshold for dynamic module selection (relative drop detection). |
| $\eta$ | 0.5 | [0.3, 0.7] | Weight balancing $C^O$ and $C^S$ in logit-viewpoint term of contribution score. |
| $m$ | 0.5 | [0.3, 0.7] | Weight balancing probability ($C^P$) and logit viewpoints in integrated score. |
| $r$ (LoRA rank) | 8 | [4, 16] | Low-rank dimension for injected LoRA adapters in selected modules. |
| $\alpha$ (LoRA scaling) | 16 | [8, 32] | Scaling factor for LoRA weight updates ($\Delta W = \alpha \cdot B \cdot A$). |
| $\lambda$ (KL weight) | 0.1 | [0.05, 0.2] | Regularization weight for KL-divergence in constrained editing loss. |

## D MORE EXPERIMENTAL RESULTS

### D.1 EXTENDED EXPERIMENTS ON QWEN2.5-VL AND VLKEB

In this section, we present additional experimental results on the Qwen2.5-VL and VLKEB benchmarks, covering the performance of KDKE across different MLLM backbones. These experiments not only validate our approach on the E-IC and E-VQA tasks using the Qwen2.5-VL backbone, but also extend the evaluation to the VLKEB benchmark, which tests general knowledge editing capabilities across a broader set of models.

For Qwen2.5-VL, we perform experiments on a 3B parameter version of the model. Results show that KDKE outperforms other editing methods across the E-IC and E-VQA tasks. In particular, KDKE achieves a significant improvement in both the reliability and locality of knowledge editing, demonstrating the effectiveness of our dynamic module selection and constrained adaptive editing techniques.

Additionally, we conduct experiments on the VLKEB benchmark across four popular MLLMs (BLIP2-OPT, LLaVA-v1.5, MiniGPT-4, and Qwen2.5-VL). This extended evaluation confirms that KDKE consistently delivers superior performance in terms of reliability, generalization, and locality, outperforming existing methods like VisEdit and other baseline editors.

The detailed results for Qwen2.5-VL and VLKEB experiments are summarized in Tables 5 and 6, showcasing the versatility and robustness of KDKE across different architectures and tasks.

Table 5: Comparison of KDKE on Qwen2.5-VL

| Method | E-IC | | | | | | E-VQA | | | | | |
|--------|------|------|------|------|------|---------|------|------|------|------|------|---------|
| | Rel.↑ | T-Gen.↑ | I-Gen.↑ | T-Loc.↑ | I-Loc.↑ | Average | Rel.↑ | T-Gen.↑ | I-Gen.↑ | T-Loc.↑ | I-Loc.↑ | Average |
| | Qwen2.5-VL(3B) | | | | | | | | | | | |
| FT-L | 97.25 | 95.42 | **99.75** | 44.83 | 18.92 | 71.23 (±0.72) | 99.88 | **99.85** | 98.64 | 66.37 | 35.41 | 77.90 (±0.68) |
| FT-V | 35.91 | 59.37 | 33.15 | 99.88 | 41.26 | 53.91 (±0.85) | 46.58 | 37.45 | 45.72 | 99.92 | 31.75 | 51.48 (±0.79) |
| KE | 68.47 | 63.59 | 60.84 | 95.83 | 46.12 | 66.97 (±0.64) | 66.92 | 62.41 | 65.08 | 96.75 | 44.63 | 67.56 (±0.61) |
| IKE | 97.35 | 77.63 | 82.49 | 14.22 | 2.08 | 54.75 (±0.92) | 99.87 | 90.84 | 91.76 | 12.89 | 1.95 | 59.46 (±0.88) |
| WilKE | 63.27 | 31.85 | 64.52 | 92.46 | 60.89 | 62.60 (±0.74) | 65.91 | 29.74 | 61.87 | 90.68 | 58.42 | 61.32 (±0.71) |
| SERAC | 93.67 | 96.31 | 90.85 | 99.95 | 0.52 | 76.26 (±0.81) | 90.58 | 92.07 | 88.96 | 99.97 | 0.41 | 74.40 (±0.83) |
| MEND | 64.85 | 37.26 | 35.84 | 91.92 | 56.38 | 57.25 (±0.76) | 91.87 | 89.95 | 90.79 | 95.43 | 64.28 | 86.66 (±0.65) |
| TP | 48.96 | 50.14 | 46.27 | 94.51 | 79.73 | 63.92 (±0.72) | 67.58 | 61.42 | 57.19 | 97.86 | 84.05 | 73.62 (±0.69) |
| LTE | 97.14 | 94.87 | 93.41 | 94.68 | 86.95 | 93.41 (±0.48) | 96.99 | 96.58 | 95.72 | 93.67 | 85.42 | 93.88 (±0.45) |
| VisEdit | 96.58 | **97.12** | 95.36 | **99.98** | 90.89 | 95.97 (±0.42) | 97.46 | 96.88 | 95.24 | **99.95** | 90.67 | 96.04 (±0.44) |
| **KDKE** | **100** | 93.51 | 97.89 | 95.68 | **94.33** | **96.48** (±0.38) | **100** | 93.74 | **98.57** | 95.91 | **91.84** | **96.99** (±0.41) |

Table 6: Comparison of KDKE on VLKEB.

| Method | VLKEB | | | | | |
|--------|-------|------|------|------|------|---------|
| | Rel.↑ | T-Gen.↑ | I-Gen.↑ | T-Loc.↑ | I-Loc.↑ | Average |
| | BLIP2-OPT(2.7B) | | | | | |
| FT-L | 98.15 | 81.55 | 96.80 | 86.47 | 95.10 | 91.61 (±0.15) |
| FT-V | 70.82 | 71.25 | 53.67 | 94.51 | **96.85** | 77.42 (±0.32) |
| KE | 49.51 | 57.13 | 41.38 | 98.28 | 30.56 | 55.37 (±0.28) |
| IKE | 99.13 | 79.47 | 95.27 | 12.35 | 1.85 | 57.61 (±0.41) |
| WilKE | 82.66 | 68.12 | 72.53 | 55.34 | 49.24 | 65.58 (±0.19) |
| SERAC | 94.37 | 82.83 | 92.55 | 60.16 | 19.45 | 69.87 (±0.37) |
| MEND | 44.60 | 61.91 | 45.21 | 92.52 | 84.38 | 65.72 (±1.10) |
| TP | 50.10 | 40.86 | 51.58 | 46.97 | 72.47 | 52.40 (±0.33) |
| LTE | 84.69 | 87.97 | 93.58 | 77.81 | 86.13 | 86.04 (±0.22) |
| VisEdit | 97.07 | 95.03 | 96.62 | **100** | 90.89 | 95.92 (±0.68) |
| **KDKE** | **100** | 95.93 | **99.47** | 94.50 | 90.51 | **96.08** (±0.25) |
| | LLaVA-v1.5(7B) | | | | | |
| FT-L | 94.29 | 87.00 | 92.22 | 91.16 | 91.37 | 91.21 (±0.31) |
| FT-V | 76.31 | 65.57 | 59.43 | **100** | 92.35 | 78.73 (±0.73) |
| KE | 54.31 | 50.86 | 47.13 | 94.15 | 37.16 | 56.72 (±1.27) |
| IKE | 96.54 | 85.47 | 90.06 | 17.64 | 5.43 | 59.03 (±0.98) |
| WilKE | 76.21 | 74.45 | 66.41 | 61.51 | 43.51 | 64.42 (±0.56) |
| SERAC | 89.77 | 89.11 | 87.92 | 66.68 | 14.20 | 69.54 (±0.82) |
| MEND | 50.77 | 55.70 | 51.65 | 87.93 | 90.43 | 67.30 (±1.14) |
| TP | 44.56 | 47.52 | 45.36 | 52.21 | 66.61 | 51.25 (±1.46) |
| LTE | 90.06 | 81.52 | 88.11 | 83.40 | 81.48 | 84.91 (±0.47) |
| VisEdit | 96.68 | 94.70 | 95.59 | 100 | **93.76** | 96.15 (±0.65) |
| **KDKE** | **100** | 99.07 | **99.38** | 98.44 | 89.46 | **97.27** (±0.20) |
| | MiniGPT-4(7B) | | | | | |
| FT-L | 90.13 | 91.47 | 86.88 | 85.34 | 87.56 | 88.28 (±0.35) |
| FT-V | 82.59 | 61.43 | 64.76 | 96.12 | 88.94 | 78.77 (±0.68) |
| KE | 58.73 | 46.22 | 51.58 | 89.37 | 42.14 | 57.61 (±1.29) |
| IKE | 92.67 | 88.13 | 85.39 | 23.48 | 8.76 | 59.69 (±0.74) |
| WilKE | 72.84 | 78.59 | 62.17 | 68.32 | 38.64 | 64.11 (±0.57) |
| SERAC | 94.53 | 84.76 | 82.34 | 72.15 | 11.23 | 69.00 (±0.83) |
| MEND | 45.29 | 61.18 | 47.43 | 92.56 | 84.37 | 66.17 (±1.15) |
| TP | 48.91 | 43.64 | 49.82 | 48.13 | 70.47 | 52.19 (±1.11) |
| LTE | 84.78 | 85.46 | 84.19 | 77.26 | 85.83 | 83.50 (±0.46) |
| VisEdit | 95.12 | 93.54 | 89.58 | **100** | **92.77** | 94.20 (±0.65) |
| **KDKE** | **100** | 99.42 | **97.64** | 97.31 | 90.06 | **96.89** (±0.27) |
| | Qwen2.5-VL(3B) | | | | | |
| FT-L | 91.17 | 92.43 | 90.34 | 95.31 | 89.13 | 91.68 (±0.87) |
| FT-V | 82.43 | 61.36 | 64.77 | 96.11 | 89.89 | 78.91 (±0.71) |
| KE | 61.44 | 47.34 | 53.34 | 89.26 | 42.28 | 58.73 (±1.31) |
| IKE | 92.22 | 88.88 | 84.93 | 23.76 | 8.32 | 59.62 (±0.96) |
| WilKE | 80.32 | 69.22 | 71.28 | 68.72 | 40.39 | 65.99 (±1.39) |
| SERAC | 93.19 | 83.24 | 92.05 | 60.56 | 18.52 | 69.51 (±0.85) |
| MEND | 47.32 | 61.82 | 47.76 | 92.14 | 85.23 | 66.85 (±1.18) |
| TP | 51.93 | 42.41 | 51.57 | 47.89 | 69.73 | 52.71 (±1.04) |
| LTE | 93.61 | 76.43 | 92.23 | 79.51 | 86.69 | 85.69 (±0.48) |
| VisEdit | 96.14 | 97.32 | **94.36** | **100** | 87.64 | 95.09 (±0.66) |
| **KDKE** | **100** | 99.60 | 92.34 | 95.74 | **93.21** | **96.18** (±0.28) |

## D.2 SENSITIVITY ANALYSIS ON CONTRIBUTION SCORE PARAMETERS

To further validate the robustness of the *Integrated Module Contribution Score*, we perform a sensitivity analysis on its key hyperparameters: $\eta$ (balancing normalized logit value $C^O$ and significant contribution $C^S$ in the logit-viewpoint term) and $m$ (balancing probability viewpoint $C^P$ and logit-viewpoint in the geometric mean). We vary $\eta \in \{0.3, 0.5, 0.7\}$ and $m \in \{0.3, 0.5, 0.7\}$, with de-

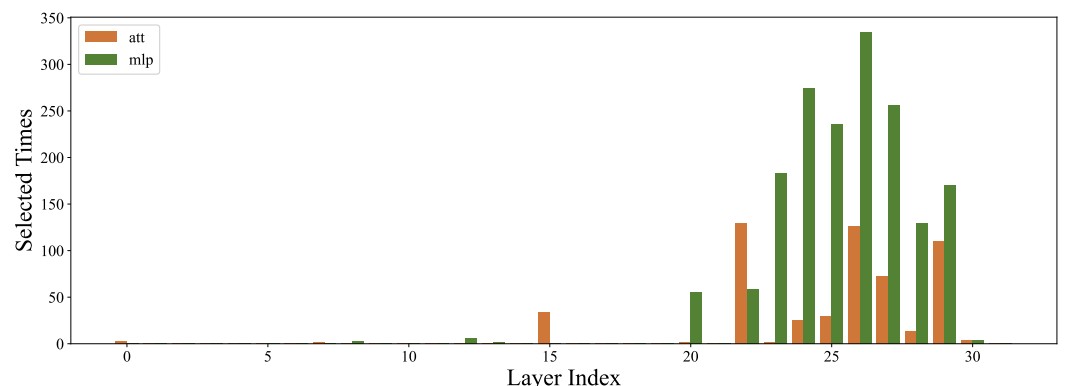

Figure 4: Module selected results on E-VQA on BLIP2-OPT

faults $\eta = 0.5$ and $m = 0.5$. Experiments follow the same setup as the main ablation: BLIP2-OPT on E-IC, evaluating the average metric score (Section 2.2).

Table 7: Sensitivity analysis of KDKE to contribution score parameters on BLIP2-OPT (E-IC).

| Parameter | $m = 0.3$ | $m = 0.5$ | $m = 0.7$ |
|---|---|---|---|
| $\eta = 0.3$ | 95.41 ($\pm$0.42) | 96.02 ($\pm$0.37) | 94.89 ($\pm$0.49) |
| $\eta = 0.5$ | 96.28 ($\pm$0.38) | **96.81** ($\pm$0.36) | 96.15 ($\pm$0.40) |
| $\eta = 0.7$ | 95.73 ($\pm$0.46) | 96.47 ($\pm$0.39) | 95.31 ($\pm$0.44) |

As illustrated in Table 7, the performance exhibits remarkable stability, with deviations below $1.2\%$ across the grid. The default configuration ($\eta = 0.5$, $m = 0.5$) achieves peak efficacy, emphasizing a balanced integration of probability and logit viewpoints. Notably, increasing $m$ toward 0.7 (favoring $C^P$) slightly reduces scores due to over-reliance on superficial probabilities, while lower $\eta$ (favoring $C^S$) enhances discriminative power but may undervalue absolute logit strength in edge cases. This analysis underscores the score's flexibility and interpretability: small perturbations do not destabilize module selection, allowing practitioners to tune for task-specific priorities (e.g., higher $m$ for probability-dominant edits). Overall, these results reinforce KDKE's parameter efficiency, minimizing the need for exhaustive tuning in real-world deployments.

### D.3 OTHER EXPERIMENTS

Following the settings in Section 4.3, we sequentially tested the performance of BLIP2-OPT on EIC and EVQA with sliding window lengths of 2 and 4, as shown in Figures 6, 7, and 8. The comprehensive results all underperformed compared to our proposed dynamic selection strategy.

## E VISUALIZING THE E-IC AND E-VQA DATASETS THOURGH EXAMPLE

To help readers unfamiliar with model editing tasks better understand the E-IC and E-VQA datasets, we provide two examples from them in Figure 9 and 10 . These examples illustrate the types of modifications and factual updates applied to the models during the editing process.

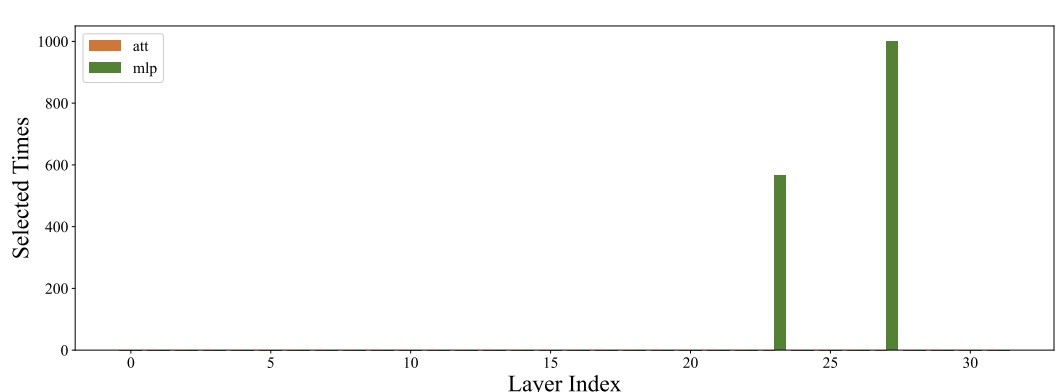

Figure 5: Module selected results on E-IC on BLIP2-OPT.

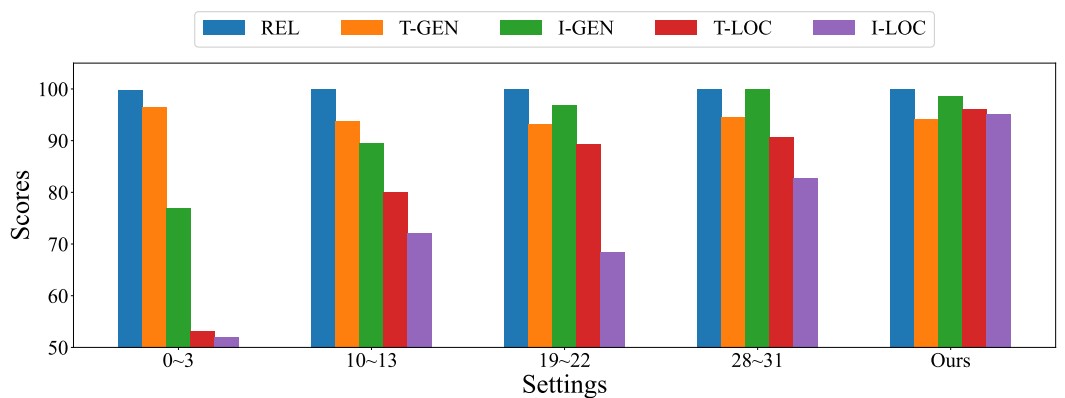

Figure 6: Ablation of Layer Select using E-IC on BLIP2-OPT with sliding window length 4.

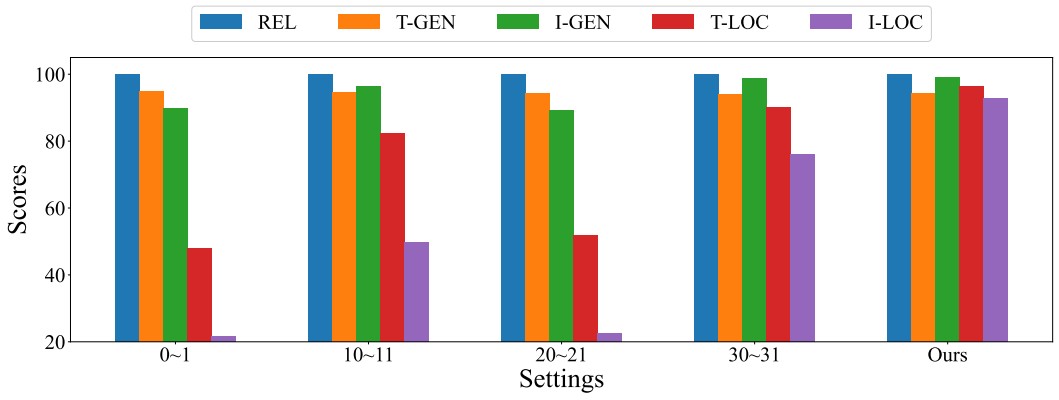

Figure 7: Ablation of Layer Select using E-VQA on BLIP2-OPT with sliding window length 2.

1134
1135
1136
1137
1138
1139
1140
1141
1142
1143
1144
1145
1146
1147
1148
1149
1150
1151
1152
1153
1154
1155
1156
1157
1158
1159
1160
1161
1162
1163
1164
1165
1166
1167
1168
1169
1170
1171
1172
1173
1174
1175
1176
1177
1178
1179
1180
1181
1182
1183
1184
1185
1186
1187

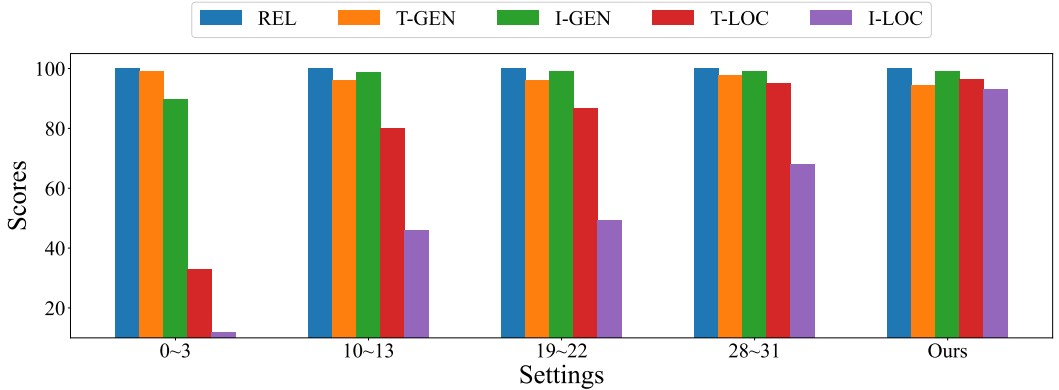

Figure 8: Ablation of Layer Select using E-VQA on BLIP2-OPT with sliding window length 4.

"src": "a photo of",
"pred": "a group of people workingin a bakery",
"rephrase":"a photograph of",
"alt": "Employees working at a cafe that has donuts on the counter.",
"image":

"image_rephrase":

"loc": "nq question: what purpose did seasonal monsoon winds have on trade",
"loc_ans": "enabled European empire expansion into the Americas and trade routes to become established across the Atlantic and Pacific oceans",
"m_loc":

"m_loc_q": "What sport can you use this for?",
"m_loc_a": "motocross"

Figure 9: A Sample of E-IC dataset.

"src": "What is the red food?",
"pred": "broccoli",
"rephrase": "What is the name of the food that is red in color?",
"alt": "tomatoes",
"image":

"image_rephrase":

"loc": "nq question: when is the next deadpool movie being released",
"loc_ans": "May 18, 2018",
"m_loc":

"m_loc_q": "What toy is this?",
"m_loc_a": "teddy bear"

Figure 10: A Sample of E-VQA dataset.

