# OpenReview forum: "Knowledge-Sensitive Dynamic Module Editing: Precise Knowledge Revision for Multimodal Large Language Models"
_ICLR.cc/2026/Conference — ICLR 2026 Conference Withdrawn Submission_

### Official Review · Reviewer_mDPs · 2025-10-28

**Soundness:** 3
**Presentation:** 2
**Contribution:** 3
**Rating:** 4
**Confidence:** 4

**Summary:**

The paper introduces the Integrated Module Contribution Score to quantify each module's contribution to the target knowledge, enabling precise identification of modules that require editing.

**Strengths:**

The paper defines the Integrated Module Contribution Score and demonstrates that it outperforms existing methods such as VisEdit and LTE on MMEdit across multiple architectures.

**Weaknesses:**

1. Lack comparison with other dynamic, layer-selective editing methods (e.g., WilKE).
2. It's easier to achieve high accuracy comparatively because of MMEdit's dataset design (generalization samples share same answers with the original question, and there is domain gap between locality samples and edited samples). It's needed to validate on more challenging datasets.
3. Further validation is required in lifelong editing scenarios.

**Questions:**

1. It's better to include comparisons of runtime and memory to identify the requirements of the module selection.
2. It's better to include the recent backbones, such as Qwen-VL 2.5 [1].
3. Complete and verify related-work citations (e.g., ICE, GRACE, MeLLo), and clarify whether GRACE belongs to context-based editing methods.
4. Perform a thorough check of symbols and grammar (e.g., quotation marks and notation).

[1] Qwen2.5-vl technical report.

---

> ### Author Response · Authors · 2025-11-20
> **Response to Weakness**
>
> Thank you for taking the time to provide such thoughtful and constructive feedback on our work.
>
> ### W1: Lack comparison with other dynamic, layer-selective editing methods (e.g., WilKE).
> As requested, we have added WilKE experiments on MMEdit across multiple MLLMs.WilKE's performance is substantially lower than expected. We believe this reflects a mismatch with the task, rather than a flaw in the method itself. Here are the key reasons:
> 1. Method mismatch in MLLMs: WilKE uses a "locate-then-edit" approach, similar to LLM methods like ROME and MEMIT. These rely on rank-one edit for token-wise parameter updates. They typically adjust the last token's internal representation in the prompt via backpropagation to guide the model toward the target output. A closed-form solution then computes the parameter changes. However, MLLM inputs are much longer than in LLMs. Editing a single token to achieve the target output raises costs and risks overfitting in the edited layers.
> 2. Dataset organization differences: As noted in our introduction, multimodal knowledge editing datasets differ from pure-text ones. Text datasets often use triplets (prompt, subject, target), where the subject is a key part of the prompt. LLM editing treats the subject's last token as the edit target. In contrast, our multimodal datasets (detailed in the appendix) replace the subject with an image. Images lack a single "central summary" token like text. To address this, we tried two approaches: (1) Using the Qwen2-VL-Chat API to manually add keywords and imitating text constraints from [1]. (2) Testing edits on multiple image token samples. Method (1) worked better, and we have updated the results using Method (1)  in the revised PDF.
>
> ### W2: Limitations of the dataset.
> Thank you for suggesting an additional dataset—it would certainly enhance breadth. However, our work already evaluates across multiple backbones and datasets , providing robust coverage. As far as we know, mainstream multimodal knowledge editing datasets all adopt this design(generalization samples share same answers with the original question).
> Regarding "the domain gap between locality samples and edited samples" raised by the reviewer, we have additionally introduced the VLKEB[2] dataset. Compared to MMEdit, VLKEB's Image Locality uses filtered similar images, significantly reducing the domain gap. The relevant experimental results have been supplemented in Appendix D.1.
>
>
> ### W3: Further validation is required in lifelong editing scenarios.
> Thank you for the reviewer's suggestion on the setting of lifelong editing, but lifelong editing is a different task setting from single editing. We would like to further clarify the task setting: KDKE is currently targeting single edit scenarios and has not been specifically designed for continuous multiple edits, such as parameter sparsity constraints, explicit "edited knowledge caching" mechanisms, etc. As mentioned by the reviewer, ROME, as well as recent representative methods such as RoseLoRA[3] and VisEdit[4], were also designed and evaluated under a single editing setting. But we also acknowledge that the reviewer made a good suggestion, and our future research goals will also focus on the setting of continuous editing.
>
> [1] Understanding Information Storage and Transfer in Multi-modal Large Language Models
>
> [2] VLKEB: A Large Vision-Language Model Knowledge Editing Benchmark
>
> [3] RoseLoRA: Row and Column-wise Sparse Low-rank Adaptation of Pre-trained Language Model for Knowledge Editing and Fine-tuning
>
> [4] Attribution Analysis Meets Model Editing: Advancing Knowledge Correction in Vision Language Models with VisEdit

---

> ### Author Response · Authors · 2025-11-26
> **Response to Questions**
>
> ### Q1: It's better to include comparisons of runtime and memory to identify the requirements of the module selection.
> As noted in your feedback, we did not compare module selection requirements in terms of runtime and memory. This is because we typically select 1-3 modules (up to a maximum of four), guided by experimental results. In MLLMs, the memory footprint per module is minimal, as shown in the table below(Using llava1.5-7b and lora rank=8 as dtype of float16 as an example)—such consumption is nearly negligible overall.
> | Module Type          | Memory Use |
> | -------------------- | ---------- |
> | One attention module | 0.25 MB    |
> | One MLP module       | 0.35 MB    |
>
> For runtime, we add no extra modules to the edited model. After parameter merging, inference time shows no significant change compared to the original model.
> ### Q2: It's better to include the recent backbones, such as Qwen-VL 2.5.
> As requested by the reviewer, we have added the experimental results of QwenVL in each dataset, as detailed in the appendix D.1
> ### Q3: Complete and verify related-work citations (e.g., ICE, GRACE, MeLLo), and clarify whether GRACE belongs to context-based editing methods.
> Thank you for point out our mistakes, it has been revised in the updated PDF.
> ### Q4: Perform a thorough check of symbols and grammar (e.g., quotation marks and notation).
> We are grateful to the reviewer for pointing this out. We have conducted a comprehensive check and grammar correction in the methodology section.

---

### Official Review · Reviewer_D35E · 2025-10-31

**Soundness:** 2
**Presentation:** 2
**Contribution:** 2
**Rating:** 4
**Confidence:** 4

**Summary:**

This paper presents KDKE, a knowledge-sensitive dynamic knowledge editing framework for multimodal large language models (MLLMs). By introducing an integrated module contribution score to quantify each module’s impact on output tokens, KDKE enables instance-wise dynamic module selection. It further employs LoRA-based constrained adaptive editing and a multi-objective loss to balance reliability, generalization, and locality. Experiments on BLIP2-OPT, LLaVA-V1.5, and MiniGPT-4 show that KDKE achieves superior performance over existing MLLM and LLM knowledge editing baselines.

**Strengths:**

1. Clear research motivation. The paper precisely identifies the key challenges in editing MLLMs—namely, the distributed and entangled nature of multimodal knowledge and the limitations of directly transferring text-only editing paradigms.
2. Comprehensive evaluation. The proposed framework is validated across multiple popular MLLMs, demonstrating broad applicability.
3. Thorough ablation studies. Each component of the proposed method is systematically analyzed through detailed ablation experiments.

**Weaknesses:**

1. Potential confusion between knowledge storage and retrieval. The key innovation, integrated module contribution score, quantifies each module’s impact on output probabilities to identify those most influential for editing. However, this may conflate *where knowledge is retrieved* with *where it is stored*. As shown in [1], such methods typically locate retrieval sites (often in the final layers) rather than true storage sites (commonly in earlier layers). By selecting components based on the last token’s contribution and applying LoRA editing, the method effectively modifies retrieval modules, potentially introducing conceptual confusion.

2. Unclear methodological description. Sections 3.1 and 3.2 lack clarity and contain multiple inconsistencies. For example, discrepancies exist between Equations (6) and (9), and the notation $c_T^p(Z)$ in line 222 is inconsistent with the rest of the text. Important symbols such as $t$ and $m$ in Equation (11) are undefined, and the variable $z$ is ambiguously used—sometimes as a token representation, other times as a module (line 259). The meaning of $\tau_{ratio}$ (line 263) is also unclear. Furthermore, Equation (11) itself is under-specified: $C_{[t]}^P(z)$ should represent a distribution, but it is unclear what raising a distribution to the $m$-th power signifies.

3. Writing and formatting issues. The paper exhibits several stylistic problems, including missing punctuation after equations throughout. It is recommended to reorganize all symbolic representations and complete methodological expressions.

4. Lack of statistical significance in results. None of the reported experiments include multiple trials or 95% confidence intervals. Statistical validation is essential for knowledge-editing tasks, as shown in [1].

5. Limited experimental exploration. While the method is tested on several models, deeper investigations are missing—for example, whether continuous edits on the same model degrade performance, how efficiency compares with other methods, and how editing quality changes under continuous edits.

[1] Locating and Editing Factual Associations in GPT.

**Questions:**

1. Is there a clear difference between how textual and visual knowledge is stored in MLLMs? Does the paper provide any deeper insight into this distinction? Understanding the modality-specific storage mechanisms is crucial, as they directly influence the most suitable strategy for knowledge editing.

2. In Dynamic Module Selection Mechanism of Section 3.2, which hyperparameters must be predefined, and how are their specific values determined in practice? Moreover, how is the rationality of these chosen values justified?

3. How stable are the editing results—does continuous editing on the same model lead to degradation in performance? Additionally, how efficient is the proposed method compared to existing approaches?

---

> ### Author Response · Authors · 2025-11-20
> **Response to Weakness**
>
> Thank you for taking the time to provide such thoughtful and constructive feedback on our work.
>
> ### W1: Potential confusion between knowledge storage and retrieval.
> We fully agree with the conclusions of ROME[1] in the context of LLMs. However, we note key differences between MLLMs and LLMs in their information flow. Existing research shows that the early layers of MLLMs mainly handle aligning and combining visual and textual features. More importantly, during generation, the output token rarely retrieves information directly from the visual token. Instead, it relies on the question token to first extract details from the visual token, and then the output token gathers context from the question token. This indirect path—from visual to question to output—means multimodal knowledge is not stored in a single "memory neuron," but is instead interconnected and spread across many parts. As a result, directly applying the LLM approach to find a specific storage site in MLLMs is neither accurate nor practical. At the same time, our ablation experiments in the Layer-wise Selection Analysis section show that editing in the early layers performs much worse than in the middle and later layers.
> ### W2: Unclear methodological description.
> Thank you for your careful reading. I have corrected the shortcomings you pointed out in the revised PDF. In addition, to help with further reading and prevent misunderstandings, let me clarify that Equation (12) does not represent a distribution. $softmax(zW_{v})$ represents a distribution, but for a specific token, $softmax(zW_{v})_{[t]}$ is a single numerical value.
> ### W3: Writing and formatting issues.
> Thank you for pointing out the writing and formatting issues. We have carefully reviewed the manuscript and made the necessary corrections, including adding punctuation after equations and reorganizing the symbolic representations for better clarity and consistency.
> ### W4: Lack of statistical significance in results.
> This is an oversight on our part. The original results were based on averages from multiple experiments, and we adopted your suggestion in the revised PDF. However, due to space constraints, we only applied statistical testing to the average results.
> ### W5: Limited experimental exploration.
> Thank you for the reviewer's suggestion on the setting of continuous editing, but continuous editing is a different task setting from single editing. We would like to further clarify the task setting: KDKE is currently targeting single edit scenarios and has not been specifically designed for continuous multiple edits, such as parameter sparsity constraints, explicit "edited knowledge caching" mechanisms, etc. As mentioned by the reviewer, ROME, as well as recent representative methods such as RoseLoRA[2] and VisEdit[3], were also designed and evaluated under a single editing setting. But we also acknowledge that the reviewer made a good suggestion, and our future research goals will also focus on the setting of continuous editing.
>
> [1] Locating and Editing Factual Associations in GPT.
>
> [2] RoseLoRA: Row and Column-wise Sparse Low-rank Adaptation of Pre-trained Language Model for Knowledge Editing and Fine-tuning
>
> [3] Attribution Analysis Meets Model Editing: Advancing Knowledge Correction in Vision Language Models with VisEdit

---

> ### Author Response · Authors · 2025-11-26
> **Response to Questions**
>
> ### Q1: Is there a clear difference between how textual and visual knowledge is stored in MLLMs? Does the paper provide any deeper insight into this distinction?
> The focus of our work is on localizing and editing at the knowledge level, rather than carefully separating and modeling storage locations for image and text modalities at the architectural level. In the MLLM we study, visual and text inputs are mapped to the same aligned multimodal representation space and combined step by step through residual streams in a shared Transformer backbone. For example, treating all image patches as separate "visual knowledge units" that need equal modeling and editing would introduce a lot of redundant information. To our knowledge, current MLLM research has no clear consensus on this issue. We appreciate the reviewer's valuable insight, but our work does not aim to provide a full, detailed model of the "modality-specific storage structure."
> ### Q2: In Dynamic Module Selection Mechanism of Section 3.2, which hyperparameters must be predefined, and how are their specific values determined in practice? Moreover, how is the rationality of these chosen values justified?
> The pre-defined hyperparameters are presented in the Appendix C of the revision PDF, and sensitivity analysis experiments have been supplemented in section 4.4 of the main text and Appendix D.2. Based on the sensitivity analysis experiments, we have selected a set of hyperparameters with relatively optimal results to apply to all datasets and MLLM.
> ### Q3: How stable are the editing results—does continuous editing on the same model lead to degradation in performance? Additionally, how efficient is the proposed method compared to existing approaches?
> Same with W5.

---

### Official Review · Reviewer_vX6Z · 2025-11-01

**Soundness:** 3
**Presentation:** 3
**Contribution:** 2
**Rating:** 4
**Confidence:** 4

**Summary:**

The paper addresses the challenge of knowledge editing in Multimodal Large Language Models (MLLMs), where traditional "locate-then-edit" methods from text-only LLMs are ineffective due to the distributed nature of multimodal knowledge. The authors propose KDKE, a Knowledge-sensitive Dynamic multimodal Knowledge Editing framework.

**Strengths:**

1.  The paper proposes the Integrated Module Contribution Score (IMCS), an attribution method designed for locating knowledge in MLLMs. It is based on the Transformer's residual stream linearity and is computed in a single forward pass, offering a potential alternative to more computationally intensive localization methods.
2.  The framework introduces a "Dynamic Module Selection" mechanism. This approach identifies knowledge-related modules on a per-instance basis, moving beyond static selection strategies. The experimental results suggest this dynamic approach is more effective than fixed-layer editing.

**Weaknesses:**

* **Ambiguity in Multi-Token Target Editing:** The paper's core metric, IMCS $C_{[t]}(z)$, is defined for a single target token $t$. This is clear for VQA tasks, but for sequence-output tasks like image captioning (E-IC), the target is a sequence. The paper fails to specify how the target token $t$ is chosen or how IMCS is aggregated in this multi-token scenario, creating a methodological ambiguity that harms reproducibility.
* **Clarity on Computational Cost:** The paper claims the IMCS calculation is "lightweight" and a "single inference step." However, it requires computing the $zW_V$ projection for all $2L$ modules, which appears significantly more expensive than a standard forward pass. A transparent analysis of this overhead (e.g., FLOPs or wall-clock time) compared to standard inference or causal tracing is missing.
* **Sensitivity of Dynamic Selection Hyperparameters:** The dynamic module selection algorithm introduces new hyperparameters ($\alpha_{gap}$, $\tau_{ratio}$) to control its thresholds. The paper provides no sensitivity analysis for these parameters, leaving it unclear how their values affect the number of selected modules and the final editing performance.
* **Under-specified Retrieval for $\mathcal{D}_{sim}$:** The locality loss relies on a set of "similar samples" $\mathcal{D}_{sim}$ from semantic retrieval. This retrieval step is a non-trivial part of the method but is not detailed. The mechanism, quality, and quantity of these retrieved samples could significantly impact locality, making this an under-specified component.

**Questions:**

1.  (Ref. Weakness 1) How is the IMCS computed for multi-token targets, such as in the E-IC dataset? Is it based on the first token, or an aggregation (e.g., mean, max, union) across the entire target sequence?
2.  (Ref. Weakness 2) Could you provide a concrete analysis of the computational overhead (e.g., FLOPs or wall-clock time) of the IMCS calculation, comparing it to both a standard forward pass and a full causal tracing run?
3.  (Ref. Weakness 3) What is the sensitivity of the model's performance and the number of selected modules to the dynamic selection hyperparameters ($\alpha_{gap}$, $\tau_{ratio}$)?
4.  (Ref. Weakness 4) Could you please detail the retrieval mechanism for the $\mathcal{D}_{sim}$ samples used in the locality loss (e.g., model, similarity metric, number of samples)?
5.  What was the motivation for freezing the LoRA 'A' matrix and only training 'B'? Did this random subspace projection prove sufficient for all edit types?

---

> ### Author Response · Authors · 2025-11-20
> **Response to Q1 and Q2**
>
> We greatly appreciate the time and effort you invested in reviewing our submission.
> ### Q1: How is the IMCS computed for multi-token targets, such as in the E-IC dataset? Is it based on the first token, or an aggregation (e.g., mean, max, union) across the entire target sequence?
> In multi-token scenarios like E-IC, we use only the first token of the sequence as the representative target token $t$ for IMCS computation. MLLM adopts an autoregressive paradigm: the generation process begins with the first token, which determines the semantic direction and knowledge activation path of the entire sequence. The first token essentially serves as the "entry point" for knowledge editing, capturing the earliest integration point of distributed knowledge in the model (as described in Section 2.1 of the paper regarding residual flow linearity). Subsequent tokens rely on this anchor point for iterative generation, so focusing on the first token enables efficient localization of core knowledge rather than dispersing attention across the entire sequence. The tail of the sequence often contains modifiers—these are highly generalizable linguistic patterns rather than specific factual knowledge. Selecting the first token filters out tail noise, ensuring more precise dynamic module selection per instance.
> ### Q2: Could you provide a concrete analysis of the computational overhead (e.g., FLOPs or wall-clock time) of the IMCS calculation, comparing it to both a standard forward pass and a full causal tracing run?
> Your observation is correct. The single inference mentioned in our article refers to a standalone process compared to traditional causal tracking methods, as these methods also involve additional computations such as noise injection or restoration during the reasoning process. In LLMs, the principle of causal tracking is destruction followed by restoration. For a given input prompt, the model generates the expected output normally, recording this "clean output." Then, noise is added to the main token range to simulate knowledge interference, which reduces the output probability. In a batch of fully noisy inputs, a hook function restores the clean hidden states at specified positions (token t, layer l), and then runs the forward pass to observe how much the logits softmax probability of the answer token recovers. Layers with higher recovery levels are considered editable layers. This method is more powerful than IMCS in terms of precision, resembling a "brute-force search." The number of inferences can be roughly estimated as: $\(length_{input} * Num_{layers} + 2\)$ (one clean output and one fully noisy output). In LLM knowledge editing, the average number of inferences is around 500, as input tokens are very short, and a single question typically contains fewer than 20 tokens. However, in MLLMs like LLaVA, images alone consist of 576 tokens, leading to tens of thousands of inferences. In our proposed dynamic process, this time cost is intolerable. The comparison of runtime between causal tracking and ours are presented in the table below.
> | **Method** | **Runtime (s)** |
> | --- | ---: |
> | Causal trace | >200 |
> | Ours | 0.13 |

---

> ### Author Response · Authors · 2025-11-20
> **Response to Q3~Q5**
>
> Thanks for reviewing our work again.
> ### Q3: What is the sensitivity of the model's performance and the number of selected modules to the dynamic selection hyperparameters($\alpha_{\text{gap}}$, $\tau_{\text{ratio}}$)
> $\alpha_{\text{gap}}$ and $\tau_{\text{ratio}}$ form a dual insurance screening mechanism that measures the absolute and relative differences between different modules, respectively. This prevents the selection of too many modules, which could cause our algorithm to degrade toward full-parameter LoRA fine-tuning. In other words, the larger the values of $\alpha_{\text{gap}}$ and $\tau_{\text{ratio}}$, the fewer modules are selected. Within a reasonable range, larger values make the mechanism more inclined to select just one candidate module for editing at a time, while smaller values make it more likely to choose 2 to 3 candidate modules. Only partial experimental results are listed here for illustration(Blip2OPT,E-IC).
> | **Config**                  | **Rel. (%)** | **T-Gen. (%)** | **I-Gen. (%)** | **T-Loc. (%)** | **I-Loc. (%)** | **Average (%)** |
> |-----------------------------|--------------|----------------|----------------|----------------|----------------|-----------------|
> | $\alpha_{\text{gap}}=0.5$, $\tau_{\text{ratio}}=0.1$ | 100         | 97.23          | 98.74          | 92.31          | 87.87          | 95.23           |
> | $\alpha_{\text{gap}}=1.5$, $\tau_{\text{ratio}}=1.0$ | 100         | 91.56          | 88.67          | 99.01          | 96.36          | 95.12           |
>
> When there are multiple candidate modules, the model's generalization will relatively increase while its locality will relatively decrease. When there are fewer candidate modules, the opposite will occur. However, both reliability and average performance remain at a high level, indicating robust parameter sensitivity.We have also added content on parameter sensitivity analysis in section 4.4 of the main text and Appendix D.2, which can be viewed in the revised PDF.
> ### Q4: Could you please detail the retrieval mechanism for the $\mathcal{D}_{\text{sim}}$ samples used in the locality loss (e.g., model, similarity metric, number of samples)?
> The data source comes from the text locality and image locality data in the training set that I used for the dataset. Using these data ensures that they are both verified and safe data with a 100% accuracy rate, and do not overlap with our edited data. Using all-MILM-L6-v2 to obtain the text embeddings of these data, a simple cosine similarity was used for similarity measurement, with a data volume of over 10000 pieces each for pure text and multimodal data.
> ### Q5: What was the motivation for freezing the LoRA 'A' matrix and only training 'B'? Did this random subspace projection prove sufficient for all edit types?
> The original intention of this design was to save resource consumption, and previous studies[1] have observed that freezing matrix A in LoRA can usually achieve performance comparable to training them. At the same time, because our editing task is a single step editing and the task is not very difficult, overfitting is very likely to occur. Although freezing the A matrix may come with a sacrifice in accuracy, using traditional lora fine-tuning can lead to a significant reduction in locality.
> Our article did not delve into the topic of 'Did this random subspace projection provide sufficient for all edit types', based on the current experimental situation, it does.
>
> [1] Lottery ticket adaptation: Mitigating destructive interference in llms.

---

### Note · Authors · 2026-01-05

I have read and agree with the venue's withdrawal policy on behalf of myself and my co-authors.